EMBO
Molecular Medicine

# GDF15 is a heart-derived hormone that regulates body growth

Ting Wang[1,2,†,‡] ⓘD, Jian Liu[1,2,†], Caitlin McDonald[1,2], Katherine Lupino[1,2], Xiandun Zhai[1,2,3], Benjamin J Wilkins[2,4], Hakon Hakonarson[5,6] & Liming Pei[1,2,4,*] ⓘD

## Abstract

The endocrine system is crucial for maintaining whole-body home-ostasis. Little is known regarding endocrine hormones secreted by the heart other than atrial/brain natriuretic peptides discovered over 30 years ago. Here, we identify growth differentiation factor 15 (GDF15) as a heart-derived hormone that regulates body growth. We show that pediatric heart disease induces GDF15 synthesis and secretion by cardiomyocytes. Circulating GDF15 in turn acts on the liver to inhibit growth hormone (GH) signaling and body growth. We demonstrate that blocking cardiomyocyte production of GDF15 normalizes circulating GDF15 level and restores liver GH signaling, establishing GDF15 as a *bona fide* heart-derived hormone that regulates pediatric body growth. Importantly, plasma GDF15 is further increased in children with concomitant heart disease and failure to thrive (FTT). Together these studies reveal a new endocrine mechanism by which the heart coordinates cardiac function and body growth. Our results also provide a potential mechanism for the well-established clini-cal observation that children with heart diseases often develop FTT.

**Keywords** body growth; failure to thrive; GDF15; heart disease; heart-derived hormone

**Subject Categories** Cardiovascular System; Metabolism

## Introduction

A central question in physiology is how different organs communi-cate with each other to maintain whole-organism homeostasis. The classical endocrine system is well documented for its essential role in inter-organ communication. In addition, research in the past 20 years has revealed that certain non-glandular organs including adipose tissue, liver, skeletal muscle, intestine, and bone have secondary endocrine functions and secrete various hormones that regulate whole-body metabolism (Zhang *et al*, 1994; Deng & Scherer, 2010; Potthoff *et al*, 2012; Stefan & Haring, 2013; Liu *et al*, 2014; Wang *et al*, 2015a; Gribble & Reimann, 2016; Karsenty & Olson, 2016). In contrast, little is known regarding heart-derived hormones besides atrial natriuretic peptide (ANP) and brain natri-uretic peptide (BNP) discovered more than 30 years ago (de Bold, 1985; Frohlich, 1985; Karsenty & Olson, 2016). Cardiac synthesis and secretion of ANP and BNP are increased in various heart diseases, and plasma level of BNP is used clinically to diagnose heart failure (Yancy *et al*, 2013). ANP and BNP in turn signal to target tissues such as kidneys and vascular smooth muscles and increase natriuresis (excretion of sodium in the urine) and decrease blood pressure, reducing the burden on the distressed heart. Identifi-cation of new heart-derived hormones and study of their biological functions will significantly advance our understanding of cardiac biology and whole-organism homeostasis.

Failure to thrive refers to poor physical growth of children, typically evaluated by height and body weight gain. FTT is seen in 5–10% of children in the U.S. primary care settings (Cole & Lanham, 2011). It is well established that pediatric heart diseases such as congenital heart disease often cause FTT, but the underly-ing mechanism is poorly understood (Menon & Poskitt, 1985; Poskitt, 1993; Forchielli *et al*, 1994; Nydegger & Bines, 2006). Intriguingly, FTT associated with pediatric heart disease often features lower circulating insulin-like growth factor 1 (IGF1) and IGF binding protein 3 (IGFBP3) levels (Barton *et al*, 1996; Dinleyici *et al*, 2007; Surmeli-Onay *et al*, 2011; Peng *et al*, 2013). GH-IGF1 signaling is a dominant mechanism regulating postnatal mammalian growth (Pilecka *et al*, 2007; Baik *et al*, 2011; Savage *et al*, 2011; Rotwein, 2012; Milman *et al*, 2016). GH secreted from the pituitary signals to the liver to stimulate the production of

1  Center for Mitochondrial and Epigenomic Medicine, Children's Hospital of Philadelphia, Philadelphia, PA, USA
2  Department of Pathology and Laboratory Medicine, Children's Hospital of Philadelphia, Philadelphia, PA, USA
3  Institute of Forensic Medicine, Henan University of Science and Technology, Luoyang, Henan, China
4  Department of Pathology and Laboratory Medicine, Perelman School of Medicine, University of Pennsylvania, Philadelphia, PA, USA
5  Center for Applied Genomics, Children's Hospital of Philadelphia, Philadelphia, PA, USA
6  Department of Pediatrics, Perelman School of Medicine, University of Pennsylvania, Philadelphia, PA, USA
   *Corresponding author. Tel: +1 267 425 2118; E-mail: lpei@mail.med.upenn.edu
   †These authors contributed equally to this work
   ‡Present address: Institute of Cell Metabolism, Shanghai Key Laboratory of Pancreatic Disease, Shanghai General Hospital, School of Medicine, Shanghai Jiaotong University, Shanghai, China

IGF1, IGFBP3, and IGFBP acid-labile subunit (IGFALS) via the JAK2-STAT5 pathway. Circulating IGF1 forms a ternary complex with IGFBP3 and IGFALS and is a major mediator of GH's effect on mammalian postnatal body growth (Czech, 1989; Pilecka *et al*, 2007; Baik *et al*, 2011; Savage *et al*, 2011; Rotwein, 2012). Mutations of almost every gene in this pathway cause severe to mild growth inhibition in humans and animals.

GDF15 (also known as MIC-1, NAG-1, PLAB, or PTGFB) is a distant member of the transforming growth factor beta (TGFβ) family of secreted proteins with pleiotropic functions (Bootcov *et al*, 1997; Yokoyama-Kobayashi *et al*, 1997; Paralkar *et al*, 1998; Bottner *et al*, 1999; Fairlie *et al*, 1999; Baek *et al*, 2001; Johnen *et al*, 2007; Unsicker *et al*, 2013). GDF15 has been shown to have a local cardio-protective role (Kempf *et al*, 2006; Xu *et al*, 2006), presumably due to its autocrine/paracrine function. Plasma GDF15 was found elevated in various heart diseases in many independent biomarker studies and in animal models (Wollert & Kempf, 2012; Baggen *et al*, 2017; Wollert *et al*, 2017). However, the organ source and biological function of increased circulating GDF15 in heart disease are little known.

Estrogen-related receptor alpha (ERRα) and gamma (ERRγ) are important transcriptional regulators of cellular metabolism especially mitochondrial functions (Huss & Kelly, 2005; Giguere, 2008; Villena & Kralli, 2008). Using cardiomyocyte-specific Myh6-Cre, we recently generated mice lacking *ERRα* and cardiac *ERRγ* (αKOγKO mice; genotype is $ERR\alpha^{-/-}ERR\gamma^{\text{flox/flox}}Myh6\text{-}Cre^+$). Control mice and those lacking *ERRα* or cardiac *ERRγ* alone exhibited normal cardiac metabolism and function, overall health and survival (Wang *et al*, 2015b). In contrast, αKOγKO mice developed lethal dilated cardiomyopathy and heart failure soon after birth (median life span of 14–15 days), featuring metabolic, contractile, and conduction dysfunctions. These results demonstrated that ERRα and ERRγ are essential for maintaining normal cardiac metabolism and function. Intriguingly, although loss of both *ERRα* and *ERRγ* occurred exclusively in cardiomyocytes (Wang *et al*, 2015b), αKOγKO mice exhibited secondary FTT as often observed in children with pediatric heart disease. Using these αKOγKO mice as a model of primary pediatric cardiomyopathy with secondary FTT, here we identify GDF15 as a heart-derived hormone that inhibits pediatric body growth. We show that GDF15 is both sufficient and required for inhibition of liver GH signaling in FTT associated with pediatric heart disease. In addition, we find that children with concomitant heart disease and FTT have elevated plasma GDF15. These results uncover a new endocrine mechanism by which the heart coordinates cardiac function and body growth. Our results also reveal a potential underlying mechanism of FTT associated with pediatric heart disease.

# Results

### Cardiac αKOγKO mice exhibit FTT with impaired liver GH signaling

Both male and female αKOγKO mice were significantly slower at gaining weight (slope of the weight curve) and were visibly smaller and shorter (height) from 5 to 7 days of age (Fig 1A and B, and data not shown), despite that weighing similar to controls around birth suggests little preterm growth defects. We hereafter chose body weight over bone or body length to determine body growth because body weight can be more accurately measured (0.01 gram precision with digital scale). Most internal organs except the heart of αKOγKO mice showed decreased absolute weight but maintained same relative weight (Appendix Fig S1A), indicating FTT-like whole-body growth inhibition rather than organ-specific developmental defects. We therefore used αKOγKO mice as a model to investigate the mechanism of FTT associated with pediatric heart disease.

The reduced body growth (about 30% less body weight) in αKOγKO mice is similar to that seen in animal and humans with defective GH-IGF1 signaling (Cui *et al*, 2007; Baik *et al*, 2011; Savage *et al*, 2011; Rotwein, 2012). We therefore measured their plasma GH and IGF1 concentrations. We found that αKOγKO mice had normal plasma GH (Fig 1C and D). In addition, expression of *Gh* in the pituitary and GH-releasing hormone (*Ghrh*) in the hypothalamus remained unchanged, indicating normal GH production and secretion (Appendix Fig S1B). Hypothalamic expression of appetite-regulating neuropeptide Y (*Npy*) and pro-opiomelanocortin (*Pomc*) was not changed either (Appendix Fig S1B). In contrast, plasma IGF1 level in αKOγKO mice was significantly decreased across multiple ages and was about 70% lower than controls by 16 days of age (Fig 1C and D). Normal plasma GH but significantly decreased IGF1 suggests that impaired liver GH signaling underlies FTT in αKOγKO mice. We therefore next examined the key components of liver GH signaling in αKOγKO mice. STAT5 phosphorylation was significantly reduced in αKOγKO mouse livers (Fig 1E and F, and Appendix Fig S1C) while upstream JAK2 phosphorylation remained unchanged (Fig 1F). Moreover, expression of STAT5 target genes *Igf1*, *Igfbp3*, and *Igfals* was all significantly decreased in αKOγKO mouse livers (Fig 1G). These result in reduced production and secretion of IGFBP3 (Fig 1H) in addition to IGF1 (Fig 1C and D). These results demonstrate that liver GH resistance underlies FTT in αKOγKO mice.

### Circulating factors mediate impaired liver GH signaling in cardiac αKOγKO mice

In our αKOγKO mouse model, loss of both *ERRα* and *ERRγ* is restricted to cardiomyocytes, which was confirmed by unchanged *ERRγ* expression in every other tissue examined (Wang *et al*, 2015b). In addition, only αKOγKO mice exhibit primary pediatric cardiomyopathy and secondary FTT, while control littermates including *ERRα* KO mice retain normal cardiac function and liver GH signaling (Fig 1C–E and G). We asked how primary cardiac genetic defects affected liver GH signaling and caused secondary FTT. We considered the possibility that the heart was communicating its functional status to the liver via nervous or endocrine mechanisms. Unfortunately, the severe cardiomyopathy and early postnatal lethality (median life span of 14–15 days) in αKOγKO mice prevented us from using surgical procedures such as vagectomy or parabiosis to investigate these possibilities. As an alternative approach to test a potential endocrine mechanism, we treated wild-type (WT) mouse primary hepatocytes with plasma from αKOγKO or control littermate αHetγWT mice. Although αKOγKO mouse plasma contained the same amount of GH (Fig 1C and D, and Appendix Fig S2), it induced significantly less STAT5 phosphorylation in WT hepatocytes (Fig 2A), recapitulating *in vivo* observations (Fig 1E and F, and Appendix Fig S1C). This suggests that the

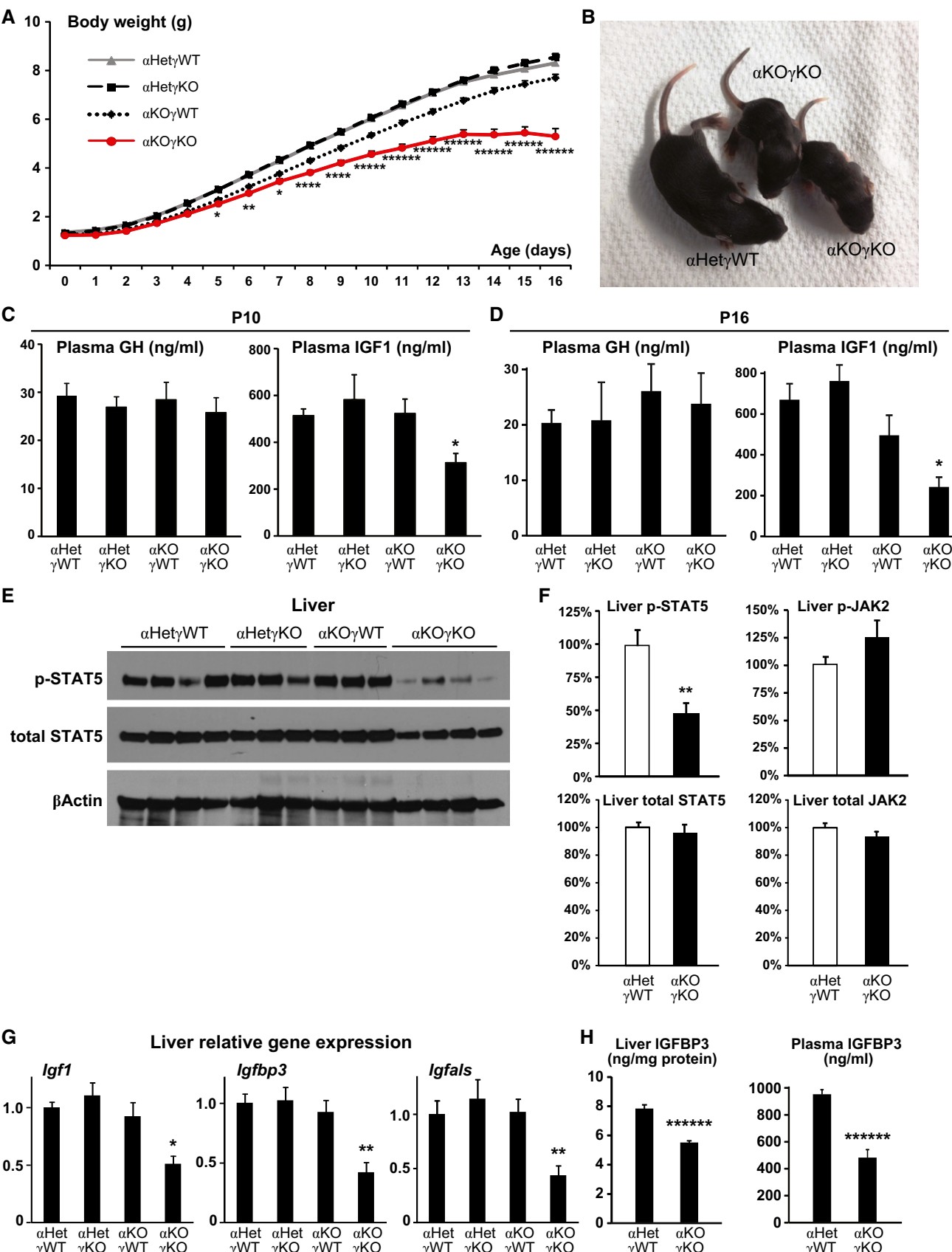

Figure 1.

◄

**Figure 1.  Cardiac αKOγKO mice exhibit FTT with impaired liver GH signaling.**

A    Daily body weight of αKOγKO and littermate control mice. *n* = 55–145 mice per group with both genders included.
B    Representative picture of 10-day-old αKOγKO and littermate control αHetγWT mice.
C, D    Plasma GH and IGF1 concentrations in 10- (C, *n* = 7–10 mice per group) and 16-day-old mice (D, *n* = 9–11 mice per group) measured by ELISA.
E    Phosphorylated (Tyr694) and total STAT5 in 10-day-old mouse livers determined by Western blot (*n* = 3–4 mice per group). β-Actin serves as a loading control.
F    Relative levels of phosphorylated and total STAT5 and JAK2 in 13-day-old mouse livers (normalized to total protein content of individual mouse liver) were quantified by ELISA (*n* = 13 mice per group).
G    Expression of STAT5 target genes *Igf1*, *Igfbp3*, and *Igfals* in 10-day-old mouse livers measured by qPCR (*n* = 7–8 mice per group).
H    Liver (normalized to total protein content, *n* = 15–16 mice per group) and plasma (*n* = 9–10 mice per group) IGFBP3 concentrations in 13-day-old mice measured by ELISA.

Data information: *$P < 0.05$, **$P < 0.01$, ****$P < 0.0001$, *****$P < 0.00001$, and ******$P < 0.000001$ between αKOγKO and all other control genotypes by *t*-test. All values are mean + s.e.m.

Source data are available online for this figure.

αKOγKO mouse plasma contains altered amount of specific factors that regulate endogenous hepatocyte GH signaling.

To determine the chemical nature of such circulating factors, we used size fractionation to separate plasma into high molecular weight (HMW, > 3 KD) and low molecular weight (LMW, < 3 KD) fractions. We then recombined different HMW and LMW fractions of the littermate control αHetγWT and αKOγKO mouse plasma to treat WT mouse hepatocytes (Fig 2B). Reconstituted αKOγKO HMW and LMW fractions recapitulated its inhibitory effect on GH signaling. Intriguingly, αKOγKO HMW and αHetγWT LMW combination also inhibited STAT5 phosphorylation (behaving as the αKOγKO plasma). In contrast, αHetγWT HMW and αKOγKO LMW combination had no effect on STAT5 phosphorylation (behaving as the αHetγWT plasma). This result suggests that the putative GH signaling regulating factors exist in the HMW fraction of αKOγKO mouse plasma and are therefore most likely proteins rather than small chemicals or cellular metabolites.

### GDF15 is a candidate heart-derived hormone that inhibits liver GH signaling and body growth

We employed two independent, unbiased strategies aiming to identify such heart-derived circulating factors that impact pediatric body growth. We used SOMAscan, an aptamer-based multiplexed proteomic platform which measures relative levels of more than 1,000 plasma proteins at one time (Gold *et al*, 2010), to identify proteins with altered plasma concentrations in the αKOγKO mice (Appendix Table S1). IGF1 appeared as the top decreased plasma protein in the SOMAscan assay, validating the power of this approach. To take account of proteins not covered by SOMAscan, we also performed RNA-Seq in αKOγKO and littermate control mouse hearts and identified genes with altered cardiac expression that encode secreted proteins (Chen *et al*, 2005; Appendix Table S2). We combined the candidates obtained from both approaches and focused on genes that showed significantly higher expression in the

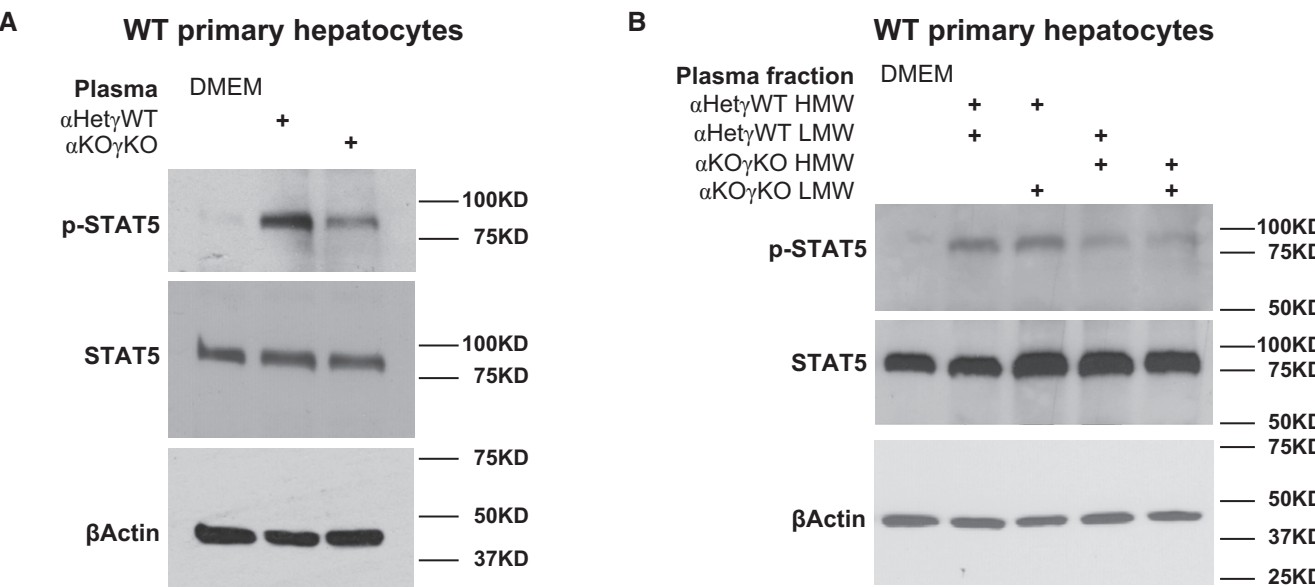

**Figure 2.  Circulating factors mediate impaired liver GH signaling in cardiac αKOγKO mice.**

A, B    Phosphorylated and total STAT5 in WT mouse primary hepatocytes treated with DMEM (control), 50% plasma in DMEM (A), or 50% plasma fractions in DMEM (B) for 1 h were determined by Western blot. β-Actin is used as loading control in all Western blots.

Source data are available online for this figure.

heart than in other tissues such as the liver (by qRT–PCR). In case of proteins with similar sequence/structure/functions such as BNP and ANP, we chose to prioritize testing one of them (e.g., BNP) first. With these criteria, we came up with eight prioritized candidates (Fig 3A). We then tested whether any of them impacted liver GH signaling in young WT mice *in vivo*, using a 96-well ELISA-based screen with plasma IGF1 as the readout. Only GDF15 significantly altered plasma IGF1 level, resulting in about 30% decrease 2 days after injection (Fig 3A). To determine whether GDF15 impacts post-natal body growth by interfering with liver GH signaling, we next performed daily GDF15 injections to WT mice from 3 days of age and monitored their daily body growth (Appendix Fig S3A). GDF15 was found to inhibit liver STAT5 phosphorylation without altering JAK2 phosphorylation (Fig 3B), decrease liver expression of STAT5 target genes *Igf1*, *Igfbp3*, and *Igfals* (Fig 3C), and reduce plasma IGF1 and IGFBP3 concentrations without affecting GH level (Fig 3D–F). Importantly, GDF15 consistently reduced body weight gain in multiple independent cohorts of WT mice as the result of this constant inhibition of liver GH signaling (Fig 3G). Individual organs such as kidneys were proportionally lighter with relative weight remaining constant (Appendix Fig S3B). GDF15 did not change hypothalamic *Npy* and *Pomc* expression (Appendix Fig S3C), suggesting that this growth-inhibiting effect is distinct from its appetite-suppressing function seen in adult mice (Johnen *et al*, 2007). This almost complete molecular and phenotypic recapitulation of the αKOγKO mice (Fig 1 and Appendix Fig S1) strongly suggests that GDF15 is a major mediator of the FTT phenotype in αKOγKO mice. In contrast, injection of BNP did not alter liver GH signaling, plasma IGF1 level, or body weight in WT mice (Appendix Fig S3D–G).

To determine whether GDF15 directly acts on the liver to alter GH signaling, we next tested whether GDF15 can inhibit GH signaling in primary hepatocytes from young WT mice. We found consistently that GDF15 inhibits hepatocyte GH signaling in a dose-dependent manner (Fig 3H). In particular, pathological concentration of GDF15 seen in the αKOγKO mouse plasma (2 ng/ml, see Fig 4C) was found to inhibit signaling of physiological level of GH based on STAT5 phosphorylation. The total cellular protein tyrosine phosphatase (PTP) activity remained unchanged (Appendix Fig S3H), suggesting that GDF15 does not affect the activity of potential PTPs known to deactivate STAT5 under these conditions (Chen *et al*, 2003; Rigacci *et al*, 2003). Together these results suggest that GDF15 directly acts on hepatocytes to inhibit GH signaling.

## GDF15 is increased in pediatric heart disease and is synthesized in cardiomyocytes

GDF15 is believed to be synthesized in the cell as a pro-protein with the C-terminal mature peptide secreted (Appendix Fig S4A; Bauskin *et al*, 2000). We found that cardiac *Gdf15* expression in αKOγKO mice was similar to control at 3 days of age, but continued to quickly rise with the development of cardiomyopathy and reached over 30-fold by 13 days of age over control mice (Fig 4A). This resulted in a significant increase of GDF15 protein in both the heart and the plasma in a similar kinetic pattern (Fig 4B–D and Appendix Fig S4A). Immunohistochemistry further showed that while completely absent in the control mouse hearts, GDF15 protein is abundant in the αKOγKO mouse hearts (Fig 4D).

Coimmunostaining with cardiomyocyte marker troponin I allowing muscle fiber visualization revealed that GDF15 was located in the cytoplasm of cardiomyocytes and did not appear to be present in any other cell types of the heart (Fig 4E).

## GDF15 is a *bona fide* heart-derived hormone that regulates liver GH signaling

These findings extend beyond the αKOγKO mouse model and have broad implications. Plasma GDF15 level was reported to be elevated in many forms of adult heart disease in both patients and animal models and was therefore recently proposed as an independent biomarker for heart diseases (Wollert & Kempf, 2012; Baggen *et al*, 2017; Wollert *et al*, 2017). However, the exact organ source and biological function of increased circulating GDF15 in heart disease remain little understood. To determine whether GDF15 is essential in mediating body growth inhibition in αKOγKO mice, we first tested this *in vitro* using GDF15 antibody to specifically deplete GDF15 in control and αKOγKO mouse plasma (Appendix Fig S4B). GDF15-depleted αKOγKO plasma largely lost its ability to inhibit GH signaling in primary hepatocytes (Appendix Fig S4C). This result suggests that GDF15 is the major GH-inhibiting factor in αKOγKO plasma. We next aimed to determine whether cardiac-derived GDF15 is critical in inhibiting liver GH signaling *in vivo*. We designed an AAV9-*Gdf15* shRNA vector to specifically knockdown *Gdf15* expression in the mouse heart (Fig 5A). Pericardial injection of AAV9 has been shown to achieve stable and relatively cardiac-specific expression of transgenes compared to other AAV serotypes (Bish *et al*, 2008). Since *Gdf15* is exclusively expressed in αKOγKO mouse cardiomyocytes (Fig 4E), we designed the AAV9 vector to ensure that *Gdf15* shRNA is solely expressed in αKOγKO mouse cardiomyocytes (Myh6-Cre[+]) but not elsewhere (Myh6-Cre[−]) (Fig 5A). The specificity of our strategy was experimentally confirmed as non-cardiac expression of *Gdf15* was not affected (Appendix Fig S4D). One week after pericardial injection of AAV9-shRNA, we found that cardiomyocyte-specific knockdown of *Gdf15* almost completely normalized cardiac *Gdf15* expression and plasma GDF15 concentration (Fig 5B and C). Importantly, the development of lethal cardiomyopathy in αKOγKO mice was little affected by cardiac *Gdf15* knockdown based on cardiac histology and cardiomyopathy marker *Bnp* expression (Fig 5D). These results provide definitive evidence that the elevated plasma GDF15 is exclusively produced by cardiomyocytes. In addition, normalization of circulating GDF15 in αKOγKO mice restored liver GH signaling (significantly increased STAT5 phosphorylation, Fig 5E) and doubled circulating IGF1 (Fig 5F) without affecting circulating GH levels (Fig 5G). Although the early lethality of αKOγKO mice (regardless of receiving control or *Gdf15* shRNA due to heart failure; Wang *et al*, 2015b) prevented us from keeping monitoring their body weight following this molecular reversal of liver GH inhibition, these results demonstrate that GDF15 is a *bona fide* heart-derived hormone that regulates liver GH signaling.

## Plasma GDF15 is elevated in children with heart disease and FTT

Last, to substantiate the clinical significance of our findings beyond animal models, we asked whether GDF15 potentially underlies clinical FTT associated with pediatric heart disease, which often features

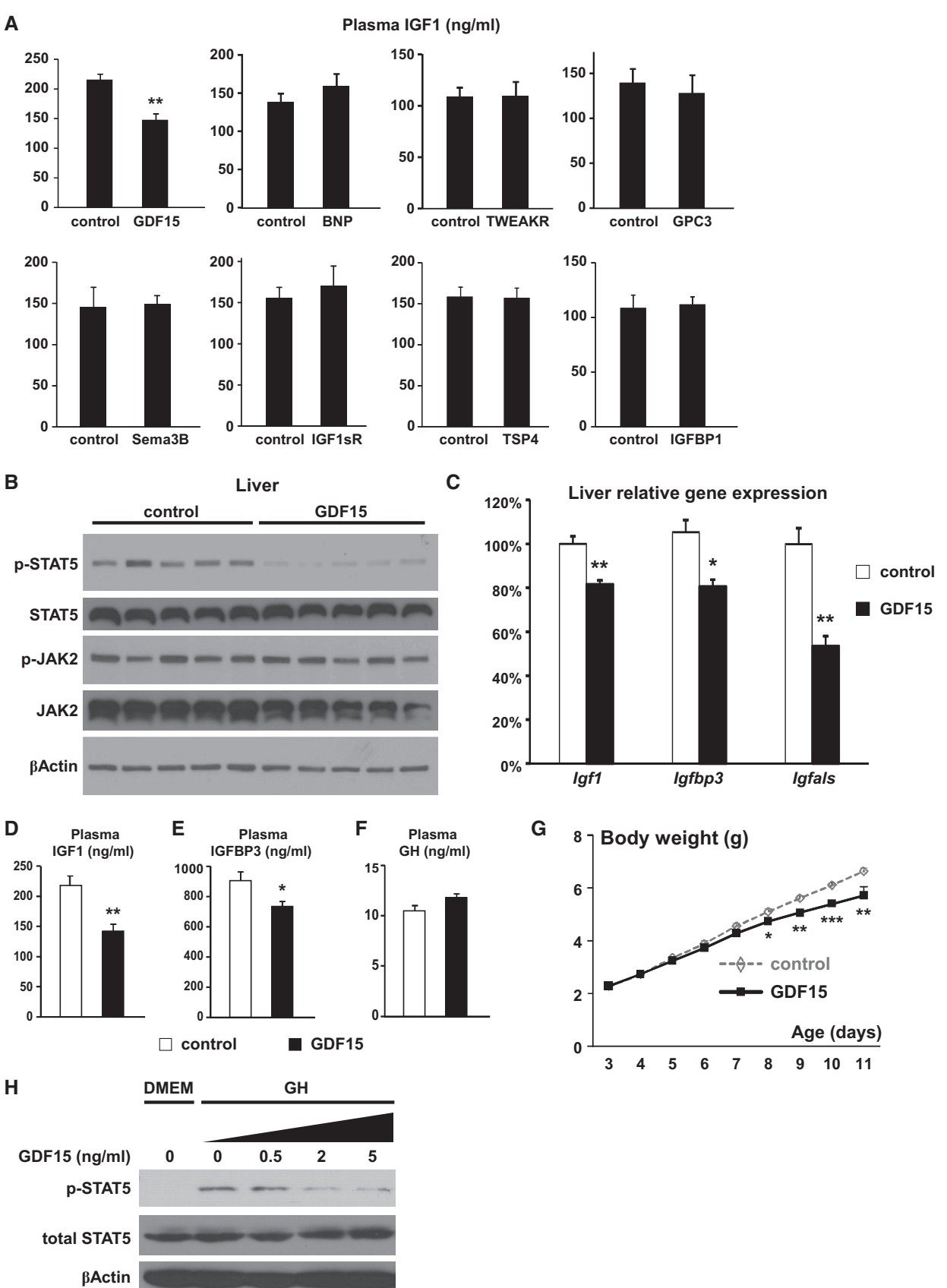

**Figure 3.**

**Figure 3.  GDF15 is a candidate heart-derived hormone that inhibits liver GH signaling and body growth.**

A  Plasma IGF1 concentrations (ng/ml) in 7-day-old weight- and gender-matched littermate WT mice injected with control or different proteins were measured by ELISA (n = 3–5 mice per group, daily i.p. injection from 5 days of age).

B–G  Liver phosphorylated and total STAT5 and JAK2 as well as β-actin (loading control) determined by Western blot (B); liver expression of *Igf1*, *Igfbp3*, and *Igfals* quantified by qPCR (C); plasma IGF1 (D), IGFBP3 (E), and GH concentrations (F) measured by ELISA; and daily body weight (G) in weight- and gender-matched littermate WT mice injected with control or GDF15 (n = 5 per group, daily i.p. injection from 3 days of age).

H  Overnight-fasted (in DMEM) WT mouse primary hepatocytes were first treated with different concentrations of GDF15 for 30 min and then with 20 ng/ml GH for 15 min. Cellular levels of phosphorylated STAT5, total STAT5, and β-actin (loading control) were determined by Western blot.

Data information: *$P < 0.05$, **$P < 0.01$, and ***$P < 0.001$ between control and GDF15 by *t*-test. All values are mean + s.e.m.
Source data are available online for this figure.

lower circulating IGF1 and IGFBP3 levels (Barton *et al*, 1996; Dinleyici *et al*, 2007; Surmeli-Onay *et al*, 2011; Peng *et al*, 2013). We measured plasma GDF15 concentrations in children diagnosed with heart disease and with either normal body weight or FTT. Both groups of children with heart disease had significantly higher plasma GDF15 levels than gender- and age-matched healthy control children (Fig 6A), consistent with previous findings that plasma GDF15 was increased in adult heart disease (Wollert & Kempf, 2012; Baggen *et al*, 2017; Wollert *et al*, 2017). Importantly, plasma GDF15 level is further significantly increased (another 80% higher) in children with concomitant heart disease and FTT than those with heart disease but normal body weight (Fig 6A), supporting elevated GDF15 as an underlying mechanism linking pediatric heart disease and FTT.

## Discussion

The rapid body growth and increased nutrients/energy demand during the pediatric period require accompanying increase of cardiac function. Here, we identified a novel endocrine mechanism by which the heart communicates with the rest of the body to coordinate body growth and cardiac function. The heart synthesizes and secretes GDF15 to inhibit body growth, thereby relieving potential extra cardiac burden as well as helping the body adapt to decreased cardiac output, a theme also seen in ANP and BNP (Fig 6B). Both GDF15 and ANP/BNP are synthesized as prohormones and processed to become active hormones. Cardiac and circulating levels of both GDF15 and ANP/BNP are highly elevated in many forms of heart disease. BNP is well established and widely used clinically for diagnosis of heart disease especially heart failure; GDF15 was also recently proposed as an independent plasma biomarker for heart diseases (Wollert & Kempf, 2012; Baggen *et al*, 2017; Wollert *et al*, 2017). Most importantly, both GDF15 and ANP/BNP are used as hormones by the heart to effect systemic changes that relieve cardiac burden: GDF15 inhibits body growth and ANP/BNP decrease blood pressure. These similarities suggest a unified endocrine mechanism that the heart exploits to coordinate cardiac function and the rest of the body (Fig 6B).

GDF15 was previously shown to mediate adult body weight loss in cachexia and obesity settings (Johnen *et al*, 2007; Tsai *et al*, 2013), through its action in the hypothalamus (reducing *Npy* and increasing *Pomc* expression) that regulates food intake. Of note body weight loss in such adult conditions predominantly impacts specific tissues (muscle and adipose tissue in cachexia; adipose tissue in obesity), with most other organs and height/body length largely unaffected. Our studies reveal a completely different and new role

of GDF15 in the pediatric period. We did not observe any changes in hypothalamic *Npy* and *Pomc* expression in either cardiac αKOγKO mice or GDF15-injected young WT mice (Appendix Figs S1B and S3C), probably because the appetite-regulating neural circuits in the hypothalamus in these young mice (1–2 weeks old) are still developing and functionally immature compared to those in adult mice (Nilsson *et al*, 2005). We find that GDF15 inhibits pediatric body growth that affects all organs (proportionally smaller and lighter in weight), rather than selectively targeting only muscle or adipose tissue. Accordingly, GDF15 functions through a distinct mechanism of inhibiting liver GH signaling that affects whole-body growth and all organs in the pediatric period.

We observed a fourfold to fivefold increase of circulating GDF15 in αKOγKO mice (Fig 4C) and similar changes in children with heart disease (Fig 6A). Such change is consistent within the range reported in human biomarker studies (Wollert & Kempf, 2012; Baggen *et al*, 2017; Wollert *et al*, 2017). Of note the normal heart produces undetectable level of GDF15 (Fig 4B and D), therefore the basal level of circulating GDF15 (Fig 4C) most likely comes from non-cardiac sources. Our cardiac-specific *Gdf15* knockdown studies suggest that cardiac-derived GDF15 in the heart disease condition contributed to all the elevated circulating GDF15 above basal level (Fig 5). This additional circulating GDF15 from the heart clearly has a significant biological impact on liver GH signaling (Fig 5). One possible explanation is that such a fourfold to fivefold increase in GDF15 concentration is enough to make a significant difference in its signaling. Another possible explanation is that heart-derived GDF15 is biochemically distinct from the basal circulating GDF15 from non-cardiac sources and potentially possesses higher biological activities. Future studies will investigate the factors that modulate GDF15 activity, and determine the detailed molecular mechanism of GDF15 signaling.

We designed AAV9-sh*Gdf15* vectors to specifically knockdown cardiomyocyte *Gdf15* in αKOγKO mice (Fig 5). In this experiment, we injected AAV-shRNA to 2-day-old mice, the age right when initial molecular changes of cardiac dysfunction were observed in cardiac αKOγKO mice (Wang *et al*, 2015b). It was expected that injected AAV9-*Gdf15* shRNA would take some time (probably in days) to be well expressed and take effect to completely block cardiac GDF15 synthesis. By this time, it is highly likely that liver GH signaling and circulating IGF1 would have already significantly decreased. We therefore anticipated that reduced liver GH signaling in αKOγKO mice would be partially reversed in the days following AAV9-sh*Gdf15* injection. We did observe that once cardiac and circulating GDF15 levels were completely normalized 1 week after AAV9-sh*Gdf15* injection, downstream liver STAT5 phosphorylation was largely restored and circulating IGF1 level was partially

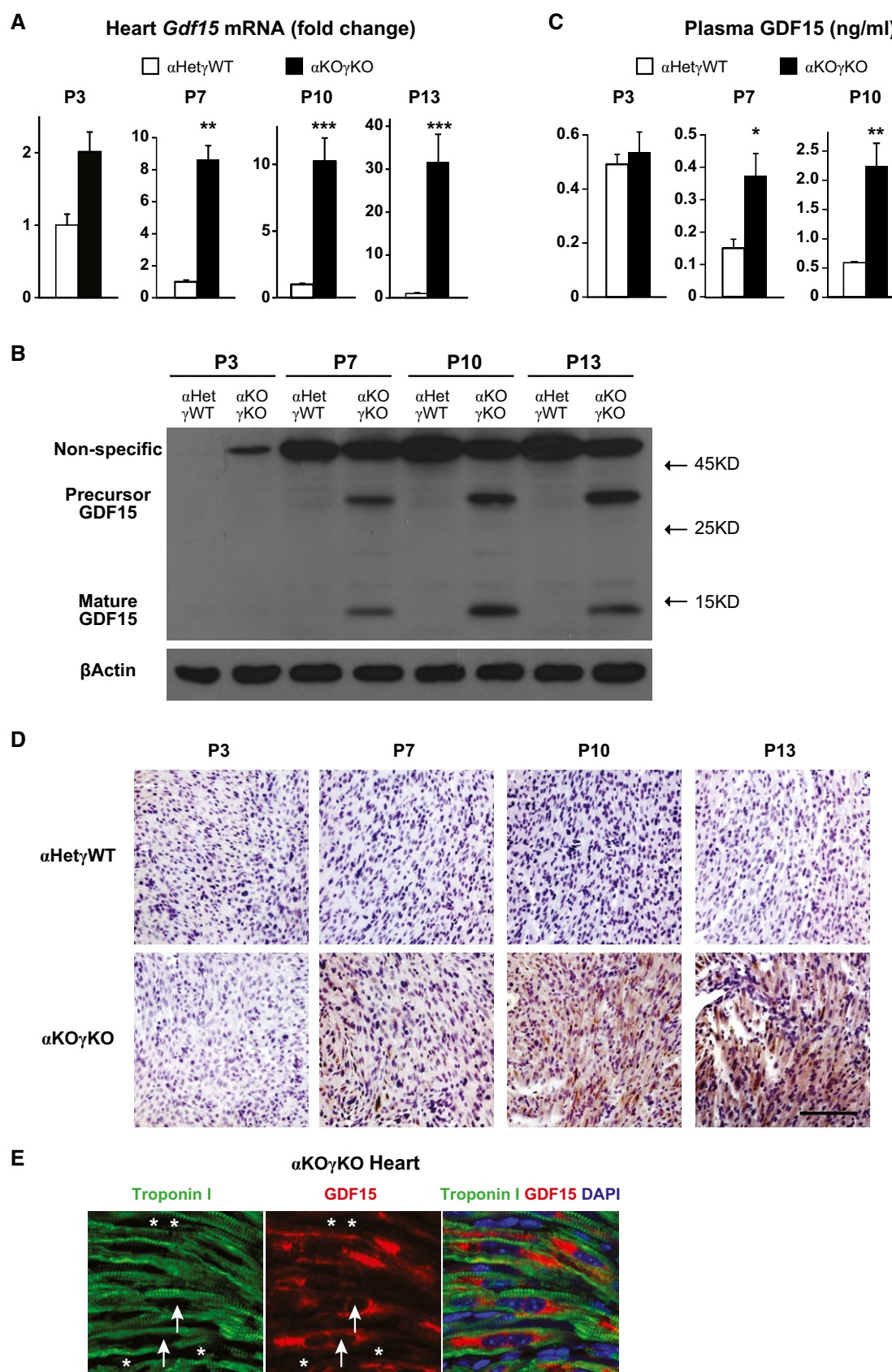

**Figure 4.**

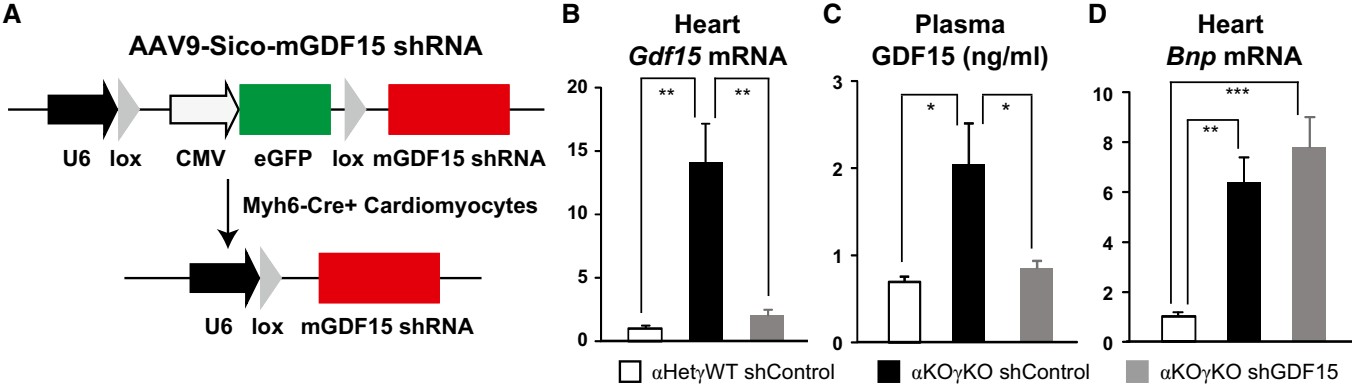

**Figure 4.  GDF15 is increased in pediatric heart disease and produced in cardiomyocytes.**

A   Expression of *Gdf15* in 3- (n = 5–7 mice per group), 7- (n = 6 mice per group), 10- (n = 8–10 mice per group), and 13-day-old (n = 12–13 mice per group) littermate mouse hearts quantified by qPCR.

B   GDF15 protein level in 3-, 7-, 10-, and 13-day-old littermate mouse hearts determined by Western blot. β-Actin serves as a loading control.

C   Plasma GDF15 concentrations in 3- (n = 5–7 mice per group), 7- (n = 6 mice per group), and 10-day-old littermate mice (n = 8–10 mice per group) measured by ELISA.

D   Representative pictures of 3-, 7-, 10-, and 13-day-old littermate αHetγWT and αKOγKO mouse heart sections stained with GDF15 antibody (brown) and counterstained with hematoxylin (purple). Scale bar: 100 μm.

E   Representative pictures of 16-day-old αKOγKO mouse hearts stained with GDF15 (red) and cardiac troponin I (green) antibodies. Arrows point to the nucleus of cardiomyocytes, and asterisks mark the nucleus of non-cardiomyocytes. Scale bar: 20 μm.

Data information: *$P < 0.05$, **$P < 0.01$, and ***$P < 0.001$ between αKOγKO and all other littermate control genotypes by *t*-test. All values are mean + s.e.m.

reversed. Although it appears that GDF15 is the major circulating factor in αKOγKO plasma that inhibits hepatocyte GH signaling *in vitro* (Appendix Fig S4B and C), it is possible that additional GDF15-independent mechanisms are involved. These include mechanisms through the nervous system that sense cardiac health and in turn regulate body growth, other heart-derived endocrine signals (including those unexamined candidates from our plasma proteomics and cardiac RNA-Seq studies), or a combination of nervous and endocrine mechanisms. Future studies in these areas will further broaden our understandings of the communication between the heart and rest of the body. Nevertheless, our current results clearly demonstrate that heart-derived GDF15 is essential for altered liver GH signaling in FTT associated with pediatric heart disease.

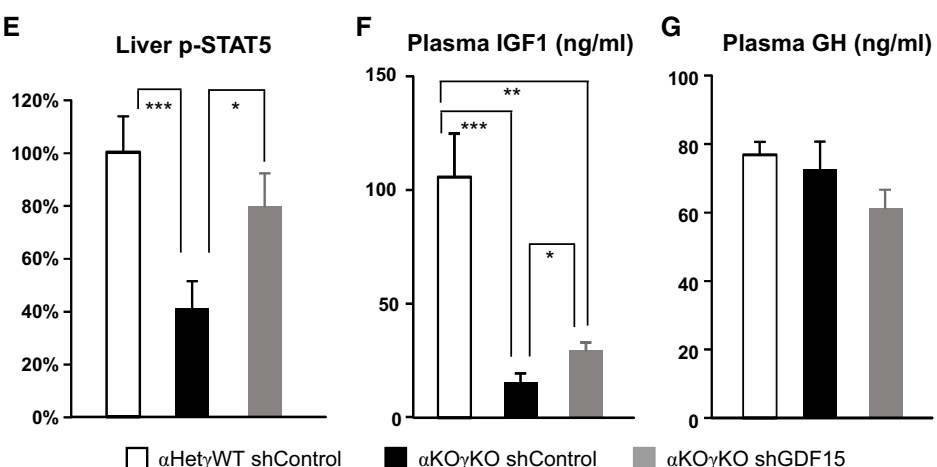

**Figure 5.  GDF15 is a *bona fide* heart-derived hormone that regulates liver GH signaling.**

A   Design of AAV9-mGDF15 shRNA construct to specifically knockdown GDF15 in Cre[+] cardiomyocytes.

B–G  Cardiac *Gdf15* expression quantified by qPCR (B), plasma GDF15 concentrations measured by ELISA (C), cardiac *Bnp* expression quantified by qPCR (D), liver phosphorylated STAT5 level measured by ELISA (E), and plasma IGF1 (F) and GH concentrations (G) measured by ELISA in 9- to 10-day-old littermate control and αKOγKO mice (n = 8–12 mice per group) that received pericardial injection of AAV9-control or *Gdf15* shRNA at 2 days of age. *$P < 0.05$, **$P < 0.01$, and ***$P < 0.001$ by *t*-test. Values are mean + s.e.m.

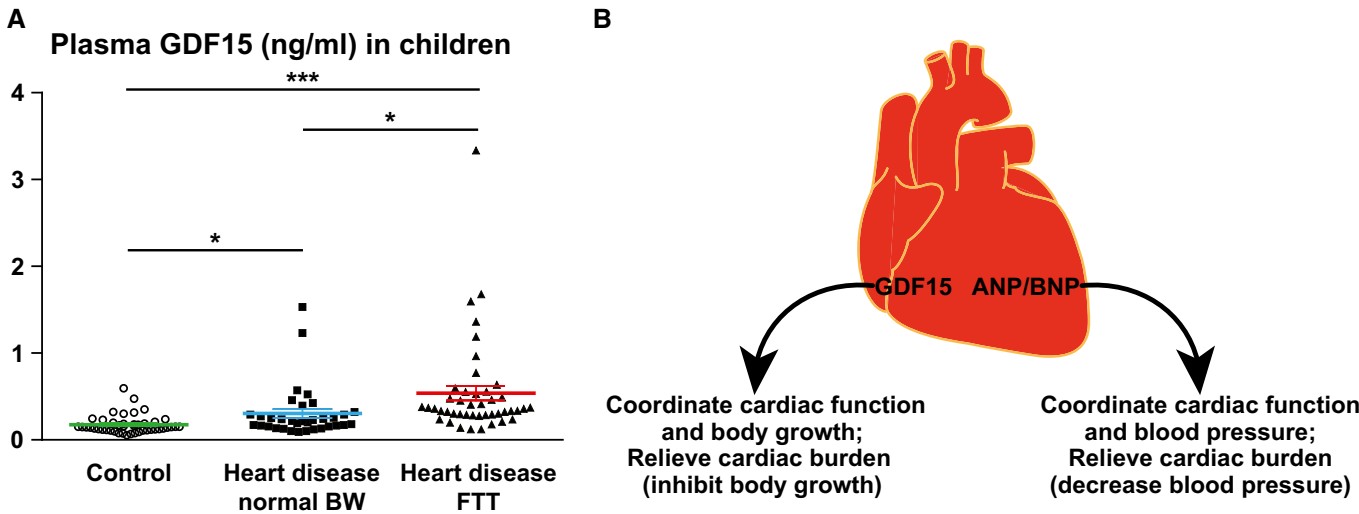

**Figure 6.  Plasma GDF15 is elevated in children with concomitant heart disease and FTT.**

A   Plasma GDF15 concentrations in 2- to 3-year-old children diagnosed with heart disease with either normal body weight (*n* = 35) or FTT (*n* = 45) and in age- and gender-matched healthy controls (*n* = 45) were measured by ELISA. *\*P* < 0.05 and *\*\*\*P* < 0.001 by *t*-test. Values are mean ± s.e.m.

B   Cartoon illustrating how GDF15 and ANP/BNP relieve cardiac burden and coordinate cardiac function with the rest of the body.

The vital function of the heart has been known for thousands of years. Besides ANP and BNP discovered over 30 years ago (de Bold, 1985; Frohlich, 1985; McGrath *et al*, 2005; Clerico *et al*, 2011; Ogawa & de Bold, 2014), only small numbers of heart-secreted factors are known (Shimano *et al*, 2012; Karsenty & Olson, 2016). However, they are not established as endocrine hormones because their known functions to date are largely limited to mere biomarkers of cardiac health or autocrine/paracrine factors that affect cardiomyocyte or cardiac fibroblast biology locally (Shimano *et al*, 2012; Karsenty & Olson, 2016). Potential heart-derived endocrine factors were suggested that impact liver and adipose tissue lipid metabolism in a few recent studies (Grueter *et al*, 2012; Baskin *et al*, 2014; Magida & Leinwand, 2014), but the exact identities of such factors remain to be determined (Karsenty & Olson, 2016). By identifying GDF15 as a new heart-derived hormone and revealing its biological function, our findings support the importance of the endocrine function of the heart and advance our understanding of the heart and its role in whole-organism homeostasis. Our studies also uncover an underlying mechanism of FTT associated with pediatric heart disease. Intriguingly, plasma GDF15 was recently reported to be increased in mitochondrial disease patients, which often features slowed body growth as well (Yatsuga *et al*, 2015; Fujita *et al*, 2016; Montero *et al*, 2016). Whether GDF15 is critical in this context remains to be determined.

## Materials and Methods

### Animal studies

All animal studies were approved by and performed under the guidelines of the Institutional Animal Care and Use Committee of the Children's Hospital of Philadelphia (CHOP). All mice were backcrossed at least six generations to and maintained in the C57BL6/J background. Mice were maintained in a temperature- and light-controlled environment with *ad libitum* access to water. Mice in holding cages (after weaning at around 28 days of age) received a standard chow diet (Lab Diet 5L0D, 58% calories from carbohydrate, 13.5% calories from fat, and 28.5% calories from proteins); nurturing moms and their pups before weaning received a breeder diet (Lab Diet 5058, 55% calories from carbohydrate, 22% calories from fat, and 23% calories from proteins). The breeding strategy to generate experimental cohorts (αHetγWT, αHetγKO, αKOγWT, and αKOγKO littermates at 1:1:1:1 ratio) was previously described (Wang *et al*, 2015b). For postnatal body growth analysis, the breeding pairs were monitored daily for birth of pups. The first day we observed new pups born was deemed as P0, and the pups were toe-clipped for identification and genotyping. We chose body weight over bone or body length to determine body growth because body weight can be more accurately measured using a digital scale with 0.01 gram precision. Mice were weighed daily between 10 am and 2 pm. Both male and female pups exhibited lethal cardiomyopathy and FTT phenotypes, and therefore, both genders were included in the study. We determined the minimum number of animals needed using power calculations based on the literature, our preliminary results and sample variation aiming to ensure 90% power. Only littermate mice were used and they were randomized to ensure that the same age, gender, and weight (mean and variation, where applicable) are represented in all groups. Investigators were blinded to the sample group allocation wherever possible. The number of mice used in each experiment and number of times experiments are replicated are described in figure legends or related Materials and Methods sections. Only mice living for at least 13 days were included in the body weight curve (Fig 1; because some αKOγKO die before this age). All tissues were collected and weighed between 12 and 4 pm to avoid the impact of circadian rhythm.

                                                                

## ELISA

Mouse blood was collected in lithium-heparin-coated microvette CB300LH (Sarstedt), and plasma was collected after spinning down at 3,000 $g$ for 5 min at 4°C. Plasma protein levels were measured using ELISA kits: mouse GH (Millipore EZRMGH-45K), mouse IGF1 (Abcam ab108874), mouse IGFBP3 (R&D MGB300), mouse GDF15 (R&D MGD150), and human GDF15 (R&D DY957). Separately, mouse livers were homogenized with kit-provided lysis buffer containing protease and phosphatase inhibitors (Roche). Phosphorylated and total STAT5 and JAK2 levels were measured using the following ELISA kits and normalized to total protein amount: p-STAT5 (Tyr694, Cell Signaling #7113C), total STAT5 (Abcam ab205714), p-JAK2 (Tyr1007/1008, ThermoFisher Scientific KH05621), and total JAK2 (Invitrogen KH05521).

## Plasma proteomics

EDTA plasma was collected from 16-day-old littermate αHetγWT and αKOγKO mice ($n = 3$ mice per group) and stored at −80°C until analysis. Measurement of relative plasma protein levels using the SOMAscan platform was performed at SomaLogic Inc. (Gold $et$ $al$, 2010).

## Gene expression analysis

We isolated total RNA from mouse tissues or cells using RNAzol RT (Molecular Research Center) following the manufacturer's instructions. We synthesized cDNA from 1 μg total RNA using iScript cDNA synthesis kit (Bio-Rad) and quantified mRNA levels by real-time qPCR using SYBR Green (Bio-Rad). We ran samples in technical triplicates and calculated relative mRNA levels using a standard curve and normalized to 36b4 mRNA level in the same sample. The qPCR primer sequences are listed in Appendix Table S3.

## RNA-Seq

Total RNA from 16-day-old littermate mouse (αHetγWT, αHetγKO, αKOγWT, and αKOγKO) hearts was extracted using RNAzol RT. We prepared two independent samples per genotype for RNA-Seq, with each sample comprised of equal amount of pooled RNA from three biological replicates (total of six littermate hearts per genotype used). Library was constructed using TrueSeq Library Prep kit (Illumina), and sequencing and bioinformatic analysis were performed by BGI@CHOP (PE100, 20 million reads). The raw and processed data are deposited in the GEO database (accession number GSE88761, https://www.ncbi.nlm.nih.gov/geo/query/acc.cgi?token = cpqnciqcrxchngl&acc = GSE88761).

## Protein analysis

For whole-cell extract preparation, cells or 20 mg of tissues was homogenized and kept in 400 μl (20×) cold RIPA buffer containing protease/phosphatase inhibitors on ice for 10 min. The homogenates were spun down at 13,000 $g$ for 10 min, and the supernatant was collected as whole-cell extract. For nuclear/cytosolic extract preparation, 20 mg of tissues was homogenized and kept in 400 μl cold buffer containing 10 mM HEPES (pH 7.4), 1.5 mM MgCl₂,

10 mM KCl, 0.5 mM DTT, and protease and phosphatase inhibitors on ice for 10 min. The homogenates were spun down at 13,000 $g$ for 10 min, and the supernatant was collected as cytosolic extract. The pellets were resuspended in 300 μl cold buffer containing 10 mM HEPES (pH 7.4), 1.5 mM MgCl₂, 420 mM NaCl, 0.2 mM EDTA, 25% glycerol, 0.5 mM DTT, and protease/phosphatase inhibitors and shaken overnight at 4°C. They were then spun down at 13,000 $g$ for 20 min, and the supernatant was collected as nuclear extract. Protein concentration was quantified by a BCA assay kit (ThermoFisher Scientific). Western blot was performed as previously described following standard procedures (Wang $et$ $al$, 2015b). Primary antibodies used were p-STAT5 (Cell Signaling #9359, 1:1,000), STAT5 (Cell Signaling #9358, 1:1,000), p-JAK2 (Cell Signaling #3776, 1:1,000), JAK2 (Millipore 06-1310, 1 μg/ml), GDF15 (Abcam ab189358, 1 μg/ml), β-actin (Cell Signaling #4970, 1:1,000), and TFIIH (Santa Cruz sc-293, 1:2,000).

## Plasma fractionation and GDF15 depletion by immunoprecipitation

Pooled plasma from 3 to 4 16-day-old αHetγWT or αKOγKO mice was used for each fractionation or GDF15 depletion experiment. Plasma was separated into high (> 3 KD) and low (< 3 KD) molecular weight fractions using Amicon Ultra 0.5-ml centrifugal 3 KD Filters (Millipore) following the manufacturer's instructions. For GDF15 depletion, plasma was incubated with 1 μg mouse GDF15 capture antibody (from ELISA kit R&D DY6385) and shaken at 4°C overnight. 250 μl dynabeads protein G (ThermoFisher Scientific 100.04D) was then added to the plasma and shaken at room temperature for 2 h. The mixture was separated using magnet, and the supernatant was collected as GDF15-depleted plasma. GDF15 concentration in plasma before and after GDF15 depletion was determined using ELISA (R&D DY6385).

## Mouse primary hepatocyte isolation and GH signaling studies

Primary hepatocytes were isolated from young WT mice as previously described (Pei $et$ $al$, 2006). Briefly, 3- to 4-week-old mouse was anesthetized and perfused at 37°C with 10 mM HEPES-buffered HBSS (1 ml/min for 5 min) and then with 0.1% type I collagenase (Worthington LS004194) in Williams E media (1 ml/min for 5 min), entering the hepatic portal vein and exiting the inferior vena cava. The perfused liver was removed and shaken to disperse cells into hepatocyte attachment media (DMEM/F12 supplemented with 0.2 mg/ml BSA, 1 mg/ml D-galactose, 0.03 mg/ml proline, 5 mM sodium pyruvate, 1× insulin-transferrin-selenium-A (ThermoFisher Scientific 51300044), 1× antibiotic-antimycotic (ThermoFisher Scientific 15240062), and 10% FBS). After mixing cells with Percoll (final concentration of Percoll is 45%) and centrifugation to collect all the viable cells, we resuspended cells in hepatocyte attachment media and plated them on collagen I-coated 24-well plates with a density of 0.2–0.3 million cells per well. 3 h later, cells were washed with DMEM to remove non-attached cells and fasted overnight in DMEM. The next morning, cells were treated with DMEM, DMEM plus 50% plasma, or 50% plasma fractions for 1 h. Alternatively, cells were first treated with different concentration of GDF15 (R&D 8146-GD/CF) for 30 min and then treated with 20 ng/ml GH (R&D 1067-GH-025/CF) for 15 min. For Western blot, cells were washed with

ice-cold PBS for three times and lysed with 60 μl lysis buffer (30 μl RIPA buffer and 30 μl 2× Laemmli sample buffer from Bio-Rad, plus protease and phosphatase inhibitors and 2.5% β-mercaptoethanol). For determining protein tyrosine phosphatase (PTP) activity, cells were washed three times with ice-cold PBS and lysed in a buffer containing 150 mM NaCl, 50 mM Tris pH 7.4, 1% NP-40, and protease inhibitors (Sorenson & Sheibani, 2002). 1 μl cell lysates (around 2 μg proteins) from each group were incubated at 37°C for 30 min with 0.2 mM PTP substrate RRLIEDAEpYAARG (Upstate 12-217) in a final volume of 50 μl of assay buffer (25 mM HEPES pH 7.2, 50 mM NaCl, 5 mM DTT, and 2.5 mM EDTA) (Kakazu *et al*, 2008). Amount of phosphate released was measured by malachite green phosphate assay kit (Cayman Chemical 10009325) following manufacturer's instructions. This was normalized to total protein, which was determined using Pierce BCA Protein Assay Kit (ThermoFisher Scientific 23225).

### Protein injection

Recombinant proteins were purchased from commercial sources: GDF15 (R&D 8146-GD/CF), IGFBP1 (R&D 1588-B1), Sema3B (R&D 5440-S3/CF), GPC3 (R&D 2119-GP/CF), TWEAKR/TNFRSF12 (R&D 1610-TW), TSP4 (R&D 7860-TH), IGF1sR (IGF1R 1-936, R&D 6630-GR/CF), and BNP 1-32 (Tocris Bioscience 3522). They were reconstituted per vendor's recommendations and further diluted in PBS for *in vivo* i.p. injection. For *in vivo* experiments, we used gender- and body weight (BW)-matched WT mice all from the same litter to exclude the impact of mom's nurturing ability on BW among litters. We repeated all short- and long-term GDF15 injections at least four times and other protein injections at least two times and obtained the same results. For short-term injection, gender- and BW-matched 5-day-old WT mice all from the same litter (*n* = 3–5 per group) were i.p. injected between 11 am and 1 pm with control (same solvent diluted in PBS) or indicated protein every 24 h, three times total. Mouse plasma was collected 1.5 h after the last injection (7 days old). The injection doses were based on literature and product information from the vendors: 500 μg/kg BW for BNP and GPC3, 100 μg/kg BW for GDF15, TWEAKR, Sema3B, TSP4, IGFBP1, and IGF1sR. For long-term injection (illustrated in Appendix Fig S3A), gender- and BW-matched (BW of all littermate pups within 10% variation) 3-day-old WT mice all from the same litter (*n* = 4–5 per group) received daily i.p. injection between 11 am and 1 pm of control, 400 μg/kg BW GDF15 or 500 μg/kg BW BNP. Plasma and tissues were collected at the end of the experiment, 1.5 h after the final injection.

### Histology

Histological studies were performed as previously described (Pei *et al*, 2011, 2015). Mice were euthanized and perfused with 3 ml PBS. Tissues were then dissected and fixed in 4% paraformaldehyde in PBS overnight. Tissues were embedded in paraffin or frozen blocks. For heart immunohistochemistry, 5–8 μm paraffin sections were used. After deparaffinization, sections were incubated in AR buffer (Vector Lab H-3300) for antigen retrieval (microwave for 5 min). Sections were then rinsed in PBST and blocked with 2% horse serum for 1 h and incubated with GDF15 (Abcam ab189358, 10 μg/ml) or troponin I (Abcam ab47003, 10 μg/ml) antibodies

overnight at 4°C. The sections were then incubated with biotin-labeled secondary antibodies (5 μg/ml) and ABC reagent (Vectastain Elite ABC Reagent RTU, Vector Lab PK-7100), and stained using peroxidase substrate solution (Vector Lab). After counterstain with hematoxylin (Vector Lab H-3401) and dehydration, the sections were mounted and imaged using a Zeiss Axio microscope. For immunofluorescence, the sections were incubated with fluorescence-labeled secondary antibodies and mounted and imaged using a Zeiss LSM710 confocal microscope.

### AAV injection

AAV9-Sico-mouse *Gdf15* shRNA (based on shAAV-260008) or scramble control shRNA was custom-built and manufactured by Vector Biolabs. Pericardial injection of $3 × 10^{11}$ genome copies AAV9 was performed in 2-day-old mice using a Hamilton syringe by adapting procedures previously described without using ultrasound guidance (Laakmann *et al*, 2013). αKOγKO mice were randomized to receive AAV9-*Gdf15* or control shRNA, and littermate control αHetγWT mice received AAV9-control shRNA. As quality control and based upon pre-established criteria, mice dead before 9 days of age or with unsuccessful cardiac *Gdf15* knockdown (presumably due to unsuccessful injection or ineffective AAV infection) were excluded from the study (2 of 10 αKOγKO mice that received AAV9-sh *Gdf15* and survived to 9–10 days of age were excluded from the study shown in Fig 5; similar success rate was observed in control groups).

### Human plasma samples

Plasma samples were from the CHOP Center for Applied Genomics biobank. Plasma samples from three groups of children were used (Appendix Table S4). Group 1: age- and gender-matched healthy controls. Group 2: 2- to 3-year-old children diagnosed with congenital heart disease and/or pediatric cardiomyopathy, with normal body weight (within 40–70% range of the standard body weight chart of the individual's age; 50% is the median body weight) at the time of sample collection. Group 3: 2- to 3-year-old children diagnosed with congenital heart disease and/or pediatric cardiomyopathy, with FTT (within 0–8.5% range of the standard body weight chart of the individual's age; most were also diagnosed with FTT) at the time of sample collection. Informed consent was obtained from all subjects and the experiments conformed to the principles set out in the WMA Declaration of Helsinki and the Department of Health and Human Services Belmont Report.

### Statistical analysis

Two-tailed distribution, two-sample unequal variance *t*-test was used to determine the statistical significance between results of two independent groups, using either Microsoft Excel (animal studies) or GraphPad Prism (human studies, allowing automatic identification and exclusion of no more than 1 outlier out of > 35 biological samples per group based on the built-in ROUT method), with $P < 0.05$ deemed as statistically significant. The *P*-values of all figures are provided in Appendix Table S5.

**Expanded View** for this article is available online.

**The paper explained**

**Problem**

Endocrine organs and the hormones they secrete regulate many essential functions in our body. The vital role of the heart has been known for thousands of years, but its function as an important endocrine organ remains little understood.

**Results**

Here, we identify a new heart-derived hormone called growth differentiation factor 15 (GDF15) that regulates body growth. GDF15 synthesis and secretion by cardiomyocytes are increased in pediatric heart disease. Circulating GDF15 in turn acts on the liver to inhibit growth hormone (GH) signaling and body growth. Blocking cardiomyocyte production of GDF15 normalizes circulating GDF15 level and restores liver GH signaling, establishing GDF15 as a *bona fide* heart-derived hormone. Plasma GDF15 is increased in children with concomitant heart disease and failure to thrive (FTT).

**Impact**

Our study advances the field of cardiac endocrinology by revealing GDF15 as a heart-derived hormone that coordinates cardiac function and body growth. These results also provide a potential mechanism for the well-established clinical observation that children with heart diseases often develop FTT.

## Acknowledgements

We thank Drs. Douglas Wallace, Mitchell Lazar, Matthew Weitzman, Amita Sehgal, Michael Marks, Mark Kahn, Jonathan Epstein, Morris Birnbaum, and Elizabeth Goldmuntz for critical discussion of the project. We thank Dr. Benjamin Prosser for showing us pericardial injection techniques. The authors and this work were supported by the Office of the Assistant Secretary of Defense for Health Affairs through the Peer Reviewed Medical Research Program under Award No. W81XWH-16-1-0400, a grant from the W.W. Smith Charitable Trust (H1407), pilot awards from the Diabetes Research Center at the University of Pennsylvania from a grant sponsored by NIH DK 19525, and NIH DK111495 (L.P.), DK099379 (B.J.W.), HG008684, and MH096891 (H.H.). X.Z. is supported by the China Scholarship Council.

## Author contributions

LP conceived and directed the project. TW, JL, and LP performed most of the experiments including independently repeated experiments. CM, KL, and XZ provided technical assistance. BJW is a pathologist who evaluated, read, and scored the histology studies. HH provided human plasma samples and contributed to the analysis and interpretation of the results. TW and LP wrote, and all authors reviewed and/or edited the manuscript.

## Conflict of interest

The authors declare that they have no conflict of interest.

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
