## [Review Process File · EMBO Molecular Medicine]

GDF15 is a heart-derived hormone that regulates body growth

Ting Wang, Jian Liu, Caitlin McDonald, Katherine Lupino, Xiandun Zhai, Benjamin J Wilkins, Hakon Hakonarson, Liming Pei

Corresponding author: Liming Pei, Children's Hospital of Philadelphia/University of Pennsylvania

Review timeline:

Submission date:	18 January 2017
Editorial Decision:	15 February 2017
Revision received:	23 April 2017
Editorial Decision:	03 May 2017
Revision received:	06 May 2017
Accepted:	11 May 2017

Transaction Report:

Editor: Céline Carret

1st Editorial Decision

15 February 2017

Thank you for the submission of your manuscript to EMBO Molecular Medicine. We have now heard back from the three referees whom we asked to evaluate your manuscript. Although the referees find the study to be of potential interest, they also raise a number of concerns that must be addressed in the next final version of your article.

You will see from the comments pasted below that while referees 1 and 2 are rather supportive of publication pending better mechanistic details on the role of GDF15 and relevant missing information and discussions are provided, referee 3 is more critical. This referee questions the experimental setting in animals (age), the clinical data (not conclusive) and would like to see a reorganised and more focused article. As all 3 referees are nevertheless in favour of inviting a revision for EMBO Molecular Medicine we would like to do so. We understand that addressing all *in vivo* experimentations requested might be too much time consuming and/or not readily possible due to high animal lethality. We would still encourage you to address as much as you can, but concentrate in the key assays that would greatly improve the conclusiveness and clinical appeal of the findings.

Revised manuscripts should be submitted within three months of a request for revision; they will otherwise be treated as new submissions, except under exceptional circumstances in which a short extension is obtained from the editor. In this case, as *in vivo* experiments are requested, let us know as soon as possible should you expect delays.

Please note that it is EMBO Molecular Medicine policy to allow only a single round of revision and that, as acceptance or rejection of the manuscript will depend on another round of review, your

responses should be as complete as possible. EMBO Molecular Medicine has a "scooping protection" policy, whereby similar findings that are published by others during review or revision are not a criterion for rejection. Should you decide to submit a revised version, I do ask that you get in touch after three months if you have not completed it, to update us on the status.

Please read below for important editorial formatting and consult our author's guidelines.

I look forward to receiving your revised manuscript.

***** Reviewer's comments *****

Referee #1 (Remarks):

In this study, $ERR\alpha$ and $ERR\gamma$ double cardiac muscle specific knockout mice are used as a model to examine how heart defects result in poor development and growth in children with heart disease and FTT. Cardiac $ERR\alpha/\gamma$ double knockout mice have reduced IGF1 levels despite normal levels of GH. Treatment of hepatocytes with plasma from $ERR\alpha/\gamma$ double knockout mice led to reduced phosphorylation of STAT5, a downstream effector of GH receptor. Unbiased approaches were used to identify GDF15 as a potential cardiac secreted protein that modulates GH signaling and IGF1 production. Treatment of mice with GDF15 reduced liver STAT5 phosphorylation, IGF1 production, and body weight. Cardiac specific knockdown of GDF15 in cardiac $ERR\alpha/\gamma$ double knockout mice partially restores GH signaling and plasma IGF1 levels. Lastly, GDF15 levels are increased in children with heart disease, and are further increased in children that fail to thrive.

GDF15 has been identified as a biomarker of various cardiac pathologies. The current study further implicates GDF15 as a cardiac-derived endocrine hormone that mediates the communication between heart and liver during cardiac pathogenesis. In this regard, the authors have provided convincing evidence to support the conclusion. How GDF15 blocks GH signaling remains unresolved, although this might be beyond the scope of the current study.

Specific Comments

1. It's unclear how GDF15 suppress STAT5 phosphorylation without affecting JAK2 phosphorylation. Fig 1E seems to show some reduction in total STAT5 protein, which was not the case in hepatocytes in Fig. 2A. The authors should include phospho- and total protein levels of STAT5 (and JAK2) in all panels to determine whether GDF15 modulates STAT5 protein stability.
2. Along the same line, the authors should treat primary hepatocytes with GH +/- GDF15 to determine its specific and direct action in the liver. One could also examine whether GDF15 increases the activity of potential phosphatases known to de-activate STAT5.
3. Do levels of GDF15 correlate with plasma IGF1 levels in Fig. 6A?

Minor Comments

1. The title should better fit the model proposed by the authors (i.e., GDF15 as a pathological modulator of GH signaling). Based on the data provided, it is unclear whether GDF15 has a normal physiological function in modulating body growth.
2. The effect of cardiac GDF15 knockdown on plasma IGF1 levels is moderate in Fig. 5F, suggesting cardiac GDF15 may not be sufficient to rescue IGF1-related phenotypes in the context of $ERR\alpha/r$ double KO. Perhaps the authors could discuss other potential mechanisms (e.g., based on the serum proteomics and/or RNA-seq results).

Referee #2 (Remarks):

Using gene-targeted mice lacking estrogen-related receptor alpha and cardiac estrogen-related receptor gamma (aKOgKO mice) as a model for congenital heart disease/failure, the authors

propose that heart-derived GDF15 reduces postnatal body growth by inhibiting growth hormone signaling in the liver. This is an interesting paper, but several questions need to be addressed.

aKOgKO mice develop lethal cardiomyopathy with a median life span of 14-15 days. Previous work has shown that GDF-15 acts as a central appetite-suppressing hormone (Johnen et al. Tumor-induced anorexia and weight loss are mediated by the TGF-beta superfamily cytokine MIC-1 (=GDF15). *Nat Med* 2007;13:1333-40). Pair-feeding is probably not possible in the first two weeks after birth, but the authors need to discuss whether the appetite-suppressant effects of GDF15 may have contributed to the observed phenotype.

Is anything known about estrogen-related receptors in congenital heart disease?

Page 4: I am not aware of any 'intracellular' functions of GDF15.

Page 4 (and elsewhere): authors should cite the relevant original works and more recent reviews on GDF15 plasma levels in cardiac disease. For example, the publication by Marin & Roldan, 2015 is just a brief comment on a paper. This paper with immediate relevance to the present work should be cited: Baggen et al. Prognostic value of N-terminal pro-B-type natriuretic peptide, troponin-T, and growth-differentiation factor 15 in adult congenital heart disease. *Circulation* 2017;135:264-79. It is not quite true that 'the organ source and biological function of increased circulating GDF15 in heart disease are unclear'. For a discussion see Wollert et al. Growth differentiation factor 15 as a biomarker in cardiovascular disease. *Clin Chem* 2017;63:140-51.

Page 9: it is not clear how the '8 top candidates' were selected. Not all of them are listed in Table EV2.

Page 20: the dosing regimens for GDF15 are not clear from the text. Please provide a supplementary figure to illustrate better.

The authors should treat hepatocytes with growth hormone +/- GDF15 in vitro to explore if GDF15 has a direct effect on GH signaling in hepatocytes.

Page 12: the authors propose that 'The heart synthesizes and secretes GDF15 to inhibit body growth, thereby relieving cardiac burden as well as helping the body adapt to decreased cardiac output'. Since neither 'cardiac burden' (e.g. afterload, blood pressure) nor cardiac output have been measured, this statement is very speculative.

Table EV4 is highly unusual. Patient characteristics (e.g. age, gender, diagnoses, NYHA class etc.) need to be presented for all groups. 'ICD codes' need to be removed.

Referee #3 (Remarks):

General comments:

This manuscript utilizes a previously reported mouse model of lethal cardiac dysfunction, the cardiomyocyte-specific estrogen-related receptor α and γ double knockout animal (henceforth ERR double KO), to demonstrate the impact of GDF15 of cardiac origin as a potential endocrine hormone causing growth failure through liver signaling mechanism. The data suggest an exciting and potentially novel role of the heart as an endocrine organ in growth. The results do demonstrate that GDF15, almost certainly of cardiac origin, does signal hepatic IGF-mediated pathways. The authors then measure plasma GDF15 levels in children with heart disease and suggest that elevated GDF15 is the mechanism underlying failure to thrive (FTT) in these children. However, there are many limitations to the experiments presented that cloud the interpretation making them insufficient to support this conclusion.

First, the manuscript is difficult to read. The huge amount of data presented includes far too much detail, minimally relevant, or completely extraneous information that detracts from the key points to be made. For example, in Figure 1, all data from the α -het/ γ KO and α KO/ γ Het are unnecessary and should be deleted. Only the data from the double KO are necessary in panels C-J. Panels C, F, and G can be deleted. Selection and presentation of the key and relevant results would allow the reader to focus on that information.

Second, because the double KO animals died at a median of "14-15 days" of age (page 7, lines 13-14 and Wang et al, 2015b) indicating severe morbidity at that age, inclusion of data from the surviving minority of 16 day old animals is problematic in terms on causative mechanisms. Double KO animals began to fall off the growth curves much earlier. Was SOMA analysis and data from, for example, more healthy, but with slowed growth, 10 and 13 day old animals obtained?

Third, the SOMA analysis (Table EV1, another example of excessive presentation of results) indicates substantial differences in more than 45 proteins increased by more than 1.5 fold in doubleKO animals versus "controls", including several that may reflect significant morbidity in these very ill animals, such as glucagon (consistent with hypoglycemia), IGFBF2, peptide YY, myoglobin (consistent with rhabdomyolysis), and ANF, and that were not tested in the primary hepatocyte assay and that may have influenced the FTT phenotype. What was the rationale for their exclusion? These other proteins may well have been those causing slowed growth, rather than just GDF15. A key experiment not reported would be to deplete the HMW double KO plasma of GDF15 only and assess impact in the p-STAT5 readout assay. Other increased intracellular and not normally secreted proteins, such as TIM14, may indicate substantial cell damage.

Fourth, the data shown in Figure 4 are compelling evidence that cardiac GDF15 mRNA expression increases dramatically by post-natal day 10 in the double KO animals at the time that growth failure is accelerating in these animals, whereas expression at post-natal day 3 is not statistically increased. The plasma GDF15 increase by day 10 is also impressive. The manuscript would be strengthened if cardiac mRNA, cardiac pro-GDF15 and mature GDF (by immunoblot and immunohistochemistry), and plasma GDF15 levels were all measured at multiple time points during days 3-14 (e.g. every other day) to demonstrate the correlation, kinetics, and time association with changes in growth.

Fifth, the shRNA knockdown results (Figure 5 and assessed in 10 day old animals) strongly support the critical conclusion that increased cardiac GDF15 synthesis and secretion do result in an endocrine effect in the liver, the exciting and key conclusion of this work. Improvement in weight gain was apparently not observed, however, a result which tempers the concept that increased cardiac GDF15 secretion alone is responsible for FTT. Because knockdown vectors were injected in 2 day old animals, the lack of impact on growth is surprising, especially in light of the authors comments (discussion, data not shown) that plasma GDF levels normalize by day 9. To adequately interpret this experiment, it is essential to know the kinetics of mRNA knockdown, GDF15 synthesis and secretion, and reduction on plasma GDF levels, results that are not provided. In addition, because the vector likely entered the circulation and not just the myocardium, systemic effects may have occurred, including knockdown of GDF15 synthesis in other tissues. No assessment of this possibility is provided. A further weakness of this experiment is that details of "intrapericardial" injection, a technically demanding or impossible task, and the success rate for actual knockdown are not given because all animals with "unsuccessful" knockdown are excluded. These data should be given.

The results in children with "heart disease" are problematic, given the lack of any description of the severity of heart disease and the huge scatter in the results, with several outliers that skew the results. The means, medians, and standard deviations are not given, and the figure does not provide the number or percentage of patients with values above two standard deviations. In fact, even the vast majority of patients with poor growth overlap with normal. That is, only 7/45 appear to exceed the normal range. This might suggest that, in fact, elevated plasma GDF15 is not associated with poor growth. Review of the ICD9 codes suggest that almost all have congenital heart disease (CHD) (codes 745, 746, 747) and not cardiomyopathy (425.4, only 3 patients among 70 total) and that the entire spectrum of anomalies, from trivial to life-threatening is included. Cardiac patients with other causes of FTT, such as chromosomal anomalies which are common in CHD, are not excluded. Therefore, the conclusion that GDF15 levels may be a useful biomarker in children cannot be justified by these data.

The discussion contains some inappropriate comments and does not focus on the key points. For example, the statement (page 12, lines 2-3), "However, pediatric heart disease results in decreased cardiac function that fails to match these increased demands" is far too broad and factually incorrect. Actually, most pediatric heart disease, especially CHD, does not cause symptoms and does not have decreased cardiac function/output.

Similarly, the statement (page 13, last two lines) "This is in contrast to most other heart disease animal models whose late-onset nature or early embryonic/neonatal lethality prohibited the chance to study the pediatric period." Is also incorrect. There are many mouse models that would allow examination in this same time frame.

These limitations detract from the key conclusions, which are reasonably well supported. First, GDF15 is synthesized in the heart, induced in this mouse model, and does serve an endocrine role to signal hepatocytes. Second, this pathway may play a role in overall growth. These are important and exciting.

However, the data do not support the authors' conclusion that cardiac GDF15 is the only factor altering growth in the mouse model and certainly not in children with heart disease.

Specific comments:

1. Introduction, page 3, line 6. A key endocrine organ to add to the list is the intestine.
2. Figure EV 1A, a "cartoon" of growth hormone signaling is superfluous, but could be part of a concluding figure to emphasize role of GDF15 in growth, e.g. Figure 6B
3. The key component of the data is hepatic phosphorylated stat-5 levels, as shown in figure 1E (10 day old) and Figure EV1, panel E (16 days).
4. All other panels in Figure EV1 are adequately described in the text and can be deleted.

1st Revision - authors' response

23 April 2017

Referee #1 (Remarks):

In this study, $ERR\alpha$ and $ERR\gamma$ double cardiac muscle specific knockout mice are used as a model to examine how heart defects result in poor development and growth in children with heart disease and FTT. Cardiac $ERR\alpha/\gamma$ double knockout mice have reduced IGF1 levels despite normal levels of GH. Treatment of hepatocytes with plasma from $ERR\alpha/\gamma$ double knockout mice led to reduced phosphorylation of STAT5, a downstream effector of GH receptor. Unbiased approaches were used to identify GDF15 as a potential cardiac secreted protein that modulates GH signaling and IGF1 production. Treatment of mice with GDF15 reduced liver STAT5 phosphorylation, IGF1 production, and body weight. Cardiac specific knockdown of GDF15 in cardiac $ERR\alpha/\gamma$ double knockout mice partially restores GH signaling and plasma IGF1 levels. Lastly, GDF15 levels are increased in children with heart disease, and are further increased in children that fail to thrive.

GDF15 has been identified as a biomarker of various cardiac pathologies. The current study further implicates GDF15 as a cardiac-derived endocrine hormone that mediates the communication between heart and liver during cardiac pathogenesis. In this regard, the authors have provided convincing evidence to support the conclusion. How GDF15 blocks GH signaling remains unresolved, although this might be beyond the scope of the current study.

Response: We thank the reviewer for his/her valuable suggestions and positive comment that we have provided convincing evidence to support the conclusion.

Specific Comments

1. It's unclear how GDF15 suppress STAT5 phosphorylation without affecting JAK2 phosphorylation. Fig 1E seems to show some reduction in total STAT5 protein, which was not the case in hepatocytes in Fig. 2A. The authors should include phospho- and total protein levels of STAT5 (and JAK2) in all panels to determine whether GDF15 modulates STAT5 protein stability.

Response: We appreciate the reviewer's point. We have included new data of phospho- and total levels of STAT5 and JAK2 (via Western blot or ELISA) in our revised manuscript as suggested. The Western blot in Fig 1E included only 3-4 mice per genotype due to PAGE gel space limit. The ELISA-based measurement allowed us to include all samples available and determine statistical significance. This new result shows that there is no significant difference in total liver STAT5, phospho JAK2 and total JAK2 in $\alpha KO\gamma KO$ mice (new Fig 1F). Similar results are seen in another age of $\alpha KO\gamma KO$ mice (new Fig EV1C) and in GDF15 treated mice (new Fig 3B).

2. Along the same line, the authors should treat primary hepatocytes with GH +/- GDF15 to determine its specific and direct action in the liver. One could also examine whether GDF15 increases the activity of potential phosphatases known to de-activate STAT5.

Response: We agree with the reviewer and performed the experiment in mouse primary hepatocytes as suggested (new Fig 3H). The results show that GDF15 inhibits GH signaling in mouse primary hepatocytes, suggesting that GDF15 directly acts on the liver.

STAT5 is known to be dephosphorylated by several protein tyrosine phosphatases (PTP) (<https://www.ncbi.nlm.nih.gov/pubmed/12615921>, <https://www.ncbi.nlm.nih.gov/pubmed/14637146>). We therefore measured PTP family phosphatases activities in mouse primary hepatocytes as suggested. Our results suggest that GDF15 does not directly change PTP activity in this experimental condition (new Fig EV3H).

3. Do levels of GDF15 correlate with plasma IGF1 levels in Fig. 6A?

Response: We thank the reviewer for this comment. Following the reviewer's suggestion, we measured plasma IGF1 concentrations in the same clinical samples. The children with both heart disease and FTT have about 10% lower average plasma IGF1 levels (2684±247 pg/ml) compared to those with heart disease and normal body weight (2978±850 pg/ml). This is consistent with our finding that GDF15 inhibits IGF1 production, although we note that this difference is not statistically significant, potentially due to small sample size and relatively large variation among samples (IGF1 is known to be sensitive to many other factors such as nutritional status, <https://www.ncbi.nlm.nih.gov/pubmed/26844335>).

Minor Comments

1. The title should better fit the model proposed by the authors (i.e., GDF15 as a pathological modulator of GH signaling). Based on the data provided, it is unclear whether GDF15 has a normal physiological function in modulating body growth.

Response: We appreciate the reviewer's comment. We respectfully point out that GDF15 can modulate WT mice body growth (Fig 3G) in normal, physiological growing conditions. Our findings support GDF15 as a heart-derived hormone based on hormone definition from both professional endocrine organizations (<http://www.endocrine.org/news-room/glossary/g-to-hypoglycemia>, <https://www.endocrinology.org/about-us/what-is-endocrinology>) and esteemed dictionaries (<https://www.merriam-webster.com/dictionary/hormone>). We therefore respectfully suggest that we keep our current title because it best summarizes the most important and novel findings of this study, that GDF15 is a heart-derived hormone and it regulates body growth.

2. The effect of cardiac GDF15 knockdown on plasma IGF1 levels is moderate in Fig. 5F, suggesting cardiac GDF15 may not be sufficient to rescue IGF1-related phenotypes in the context of ERR α /r double KO. Perhaps the authors could discuss other potential mechanisms (e.g., based on the serum proteomics and/or RNA-seq results).

Response: We thank the reviewer for the suggestion. We have included discussion of other potential mechanisms in our revised manuscript as suggested (page 15, highlighted).

Referee #2 (Remarks):

Using gene-targeted mice lacking estrogen-related receptor alpha and cardiac estrogen-related receptor gamma (aKOgKO mice) as a model for congenital heart disease/failure, the authors propose that heart-derived GDF15 reduces postnatal body growth by inhibiting growth hormone signaling in the liver. This is an interesting paper, but several questions need to be addressed.

Response: We thank the reviewer for his/her insightful suggestions and comment that our paper is interesting.

aKOgKO mice develop lethal cardiomyopathy with a median life span of 14-15 days. Previous work has shown that GDF-15 acts as a central appetite-suppressing hormone (Johnen et al. Tumor-induced anorexia and weight loss are mediated by the TGF-beta superfamily cytokine MIC-1 (=GDF15). Nat Med 2007;13:1333-40). Pair-feeding is probably not possible in the first

two weeks after birth, but the authors need to discuss whether the appetite-suppressant effects of GDF15 may have contributed to the observed phenotype.

Response: Johnen et al showed that GDF15 suppressed appetite at least partially by reducing NPY and increasing POMC expression in adult mouse hypothalamus. We measured hypothalamic NPY and POMC expression in our cardiac α KO γ KO mice and in GDF15-injected young WT mice, and observed no change (new Fig EV1B and Fig EV3C). Of note the appetite-regulating neural circuits in the hypothalamus in these young mice (1-2 weeks old) are still developing and remain functionally immature compared to those in adult mice (<https://www.ncbi.nlm.nih.gov/pubmed/15804403>). We think this is probably why we did not observe NPY and POMC expression changes in young mice upon GDF15 treatment. We therefore believe that the appetite-suppressant effects of GDF15 in adult mice are unlikely to contribute to the slow body growth of cardiac α KO γ KO mice. We have included these discussions and new data in our revised manuscript (page 13-14 highlighted, new Fig EV1B and EV3C).

Is anything known about estrogen-related receptors in congenital heart disease?

Response: Although decreased expression of ERR α and its target genes have been associated with heart failure (<https://www.ncbi.nlm.nih.gov/pubmed/19061896>), to our knowledge there are no mutations of ERR associated with human congenital heart disease. We previously demonstrated that only loss of all 4 alleles of ERR α and ERR γ results in heart failure (<http://www.ncbi.nlm.nih.gov/pubmed/25624346>), which is presumably rare.

Page 4: I am not aware of any 'intracellular' functions of GDF15.

Response: There is one recent report of possible intracellular function of GDF15 (<https://www.ncbi.nlm.nih.gov/pubmed/25893289>). We agree with the reviewer that GDF15 most likely functions in the autocrine/paracrine fashion in the context cited in page 4. We therefore removed “intracellular” in our revised manuscript.

Page 4 (and elsewhere): authors should cite the relevant original works and more recent reviews on GDF15 plasma levels in cardiac disease. For example, the publication by Marin & Roldan, 2015 is just a brief comment on a paper. This paper with immediate relevance to the present work should be cited: Baggen et al. Prognostic value of N-terminal pro-B-type natriuretic peptide, troponin-T, and growth-differentiation factor 15 in adult congenital heart disease. *Circulation* 2017;135:264-79. It is not quite true that 'the organ source and biological function of increased circulating GDF15 in heart disease are unclear'. For a discussion see Wollert et al. Growth differentiation factor 15 as a biomarker in cardiovascular disease. *Clin Chem* 2017;63:140-51.

Response: We thank the reviewer for pointing out these two references, which we have added in our revised manuscript. We apologize for the confusion regarding “the organ source and biological function of increased circulating GDF15 in heart disease are unclear”. We meant that their organ source was not rigorously determined using organ/cell type-specific GDF15 knockdown approaches as we did. The cited review (Wollert et al, *Clin Chem* 2017) stated that “So far, little is known about the tissues that produce GDF-15 in patients with CV disease”. We therefore rephrased this sentence accordingly to “the organ source and biological function of increased circulating GDF15 in heart disease are little known” (page 4, highlighted).

Page 9: it is not clear how the '8 top candidates' were selected. Not all of them are listed in Table EV2.

Response: We appreciate the reviewer’s point. The RNA-Seq data (Table EV2) included differentially expressed genes encoding secreted proteins in the ERR α / γ double KO hearts. However, this list may not account for candidates that affect the secretion or maturation of putative heart-derived factors. We therefore combined data from both RNA-Seq (Table EV2) and SOMAscan (Table EV1) studies. Because we were looking for heart-derived factors, we next narrowed the list to those genes that showed significantly higher expression in the heart than in other tissues such as the liver. In case of proteins with close sequence/structure/functions such as BNP

and ANP, we chose to prioritize testing one of them (BNP) first *in vivo*. We have provided these additional details in our revised manuscript (page 8-9).

Page 20: the dosing regimens for GDF15 are not clear from the text. Please provide a supplementary figure to illustrate better.

Response: We thank the reviewer for the suggestion. We have provided an illustration of the GDF15 dosing regimen (new Fig EV3A) in the revised manuscript as suggested.

The authors should treat hepatocytes with growth hormone +/- GDF15 *in vitro* to explore if GDF15 has a direct effect on GH signaling in hepatocytes.

Response: We agree with the reviewer and performed the experiment as suggested (new Fig 3H). The results show that GDF15 inhibits GH signaling in mouse primary hepatocytes, suggesting a direct effect. This result is included and discussed in the revised manuscript (new Fig 3H, page 9-10, highlighted).

Page 12: the authors propose that 'The heart synthesizes and secretes GDF15 to inhibit body growth, thereby relieving cardiac burden as well as helping the body adapt to decreased cardiac output'. Since neither 'cardiac burden' (e.g. afterload, blood pressure) nor cardiac output have been measured, this statement is very speculative.

Response: We thank the reviewer for the comment. We respectively point out that we have previously shown that ERR α/γ double KO mice have significantly decreased cardiac output (only 1/4 of control, measured by echocardiography, <http://www.ncbi.nlm.nih.gov/pubmed/25624346>). The small size and young age of ERR α/γ double KO mice make direct measurement of cardiac burden (such as blood pressure) technically challenging. Because this is the Discussion section we intended to offer our thoughts of the physiological rationale of GDF15 in coordinating body growth and pediatric heart function. We have revised this sentence to “The heart synthesizes and secretes GDF15 to inhibit body growth, thereby relieving potential extra cardiac burden as well as helping the body adapt to decreased cardiac output” following the reviewer’s suggestion (page 13, highlighted).

Table EV4 is highly unusual. Patient characteristics (e.g. age, gender, diagnoses, NYHA class etc.) need to be presented for all groups. 'ICD codes' need to be removed.

Response: We apologize for this oversight. We have revised Table EV4 following the reviewer’s suggestion and included information of age, gender and diagnosis in our revised manuscript. We apologize that NYHA codes for these samples are not available (most were collected a while ago), but we have included the actual underlying heart disease diagnosis for each of the child in the revised Table EV4 and hope this is acceptable.

Referee #3 (Remarks):

General comments:

This manuscript utilizes a previously reported mouse model of lethal cardiac dysfunction, the cardiomyocyte-specific estrogen-related receptor α and γ double knockout animal (henceforth ERR double KO), to demonstrate the impact of GDF15 of cardiac origin as a potential endocrine hormone causing growth failure through liver signaling mechanism. The data suggest an exciting and potentially novel role of the heart as an endocrine organ in growth. The results do demonstrate that GDF15, almost certainly of cardiac origin, does signal hepatic IGF-mediated pathways. The authors then measure plasma GDF15 levels in children with heart disease and suggest that elevated GDF15 is the mechanism underlying failure to thrive (FTT) in these children. However, there are many limitations to the experiments presented that cloud the interpretation making them insufficient to support this conclusion.

Response: We thank the reviewer for his/her valuable suggestions and comment that our data suggest an exciting and potentially novel role of the heart as an endocrine organ in growth.

First, the manuscript is difficult to read. The huge amount of data presented includes far too much detail, minimally relevant, or completely extraneous information that detracts from the key points to be made. For example, in Figure 1, all data from the α -het/ γ KO and α KO/ γ Het are unnecessary and should be deleted. Only the data from the double KO are necessary in panels C-J. Panels C, F, and G can be deleted. Selection and presentation of the key and relevant results would allow the reader to focus on that information.

Response: We appreciate the suggestions and have accordingly reorganized the data. We removed as many data as possible as suggested (except those requested by other reviewers). We respectfully point out that the livers of ERR α / γ double KO mice are genotypically ERR α KO, so it is essential to include α KO/ γ WT mice as controls in liver studies to exclude the impact of different liver genetic backgrounds, at least when certain data (weight, gene expression, etc) were presented for the first time. We followed the reviewer's suggestion and made every effort to limit presenting these controls to minimum unless absolutely necessary (removed from later figures).

Second, because the double KO animals died at a median of "14-15 days" of age (page 7, lines 13-14 and Wang et al, 2015b) indicating severe morbidity at that age, inclusion of data from the surviving minority of 16 day old animals is problematic in terms on causative mechanisms. Double KO animals began to fall off the growth curves much earlier. Was SOMA analysis and data from, for example, more healthy, but with slowed growth, 10 and 13 day old animals obtained?

Response: We thank the reviewer for the comment. We used plasma from 16 days old mice due to technical reasons. The SOMAscan assay requires at least 75 μ l plasma per sample. Only older mice allowed collection of such volume of plasma. We were very careful and made sure those 16-day-old mice used for SOMAscan analysis were not losing weight at the time (still growing but gaining less weight than controls).

Third, the SOMA analysis (Table EV1, another example of excessive presentation of results) indicates substantial differences in more than 45 proteins increased by more than 1.5 fold in double KO animals versus "controls", including several that may reflect significant morbidity in these very ill animals, such as glucagon (consistent with hypoglycemia), IGFBB2, peptide YY, myoglobin (consistent with rhabdomyolysis), and ANF, and that were not tested in the primary hepatocyte assay and that may have influenced the FTT phenotype. What was the rationale for their exclusion? These other proteins may well have been those causing slowed growth, rather than just GDF15. A key experiment not reported would be to deplete the HMW double KO plasma of GDF15 only and assess impact in the p-STAT5 readout assay. Other increased intracellular and not normally secreted proteins, such as TIM14, may indicate substantial cell damage.

Response: We thank the reviewer for the comment. We have provided additional details in the revised manuscript regarding the rationale of the selection of candidate proteins (page 8-9). Change in plasma TIM14 of cardiac ERR α / γ double KO mice was not statistically significant (Table EV1). We did not test IGFBP2 and ANF because we already tested IGFBP1 and BNP (Fig 3A), proteins with close sequence/structure/functions that exhibited even greater changes in cardiac ERR α / γ double KO mice. We did not test glucagon and peptide YY because they were barely expressed in the hearts of ERR α / γ double KO mice. Following the reviewer's suggestions we performed experiments which show that peptide YY and myoglobin does not impact GH signaling in mouse primary hepatocytes. These data are presented below for the reviewer, to keep the paper focused and concise as suggested by the reviewer.

We agree with the reviewer that it is important to determine whether depletion of GDF15 from $ERR\alpha/\gamma$ double KO plasma would reduce GH signaling. We demonstrated that we can use a GDF15 antibody to specifically deplete GDF15 in $ERR\alpha/\gamma$ double KO plasma, and this GDF15-depleted $ERR\alpha/\gamma$ double KO plasma largely lost its ability to inhibit GH signaling (new Fig EV4B and C). These new data strongly support the conclusion that GDF15 is a major GH-inhibiting factor in $ERR\alpha/\gamma$ double KO plasma.

Fourth, the data shown in Figure 4 are compelling evidence that cardiac GDF15 mRNA expression increases dramatically by post-natal day 10 in the double KO animals at the time that growth failure is accelerating in these animals, whereas expression at post-natal day 3 is not statistically increased. The plasma GDF15 increase by day 10 is also impressive. The manuscript would be strengthened if cardiac mRNA, cardiac pro-GDF15 and mature GDF (by immunoblot and immunohistochemistry), and plasma GDF15 levels were all measured at multiple time points during days 3-14 (e.g. every other day) to demonstrate the correlation, kinetics, and time association with changes in growth.

Response: We agree with the reviewer that such kinetic studies would strengthen the current manuscript, and we have now generated enough mice of additional ages and performed the suggested experiments. These new data from multiple ages (about every 3 days) are now included in the revised manuscript (new Fig 4). The results show that the time course of increased heart-derived GDF15 strongly correlates with the body growth inhibition observed in $ERR\alpha/\gamma$ double KO mice.

Fifth, the shRNA knockdown results (Figure 5 and assessed in 10 day old animals) strongly support the critical conclusion that increased cardiac GDF15 synthesis and secretion do result in an endocrine effect in the liver, the exciting and key conclusion of this work. Improvement in weight gain was apparently not observed, however, a result which tempers the concept that increased cardiac GDF15 secretion alone is responsible for FTT. Because knockdown vectors were injected in 2 day old animals, the lack of impact on growth is surprising, especially in light of the authors comments (discussion, data not shown) that plasma GDF levels normalize by day 9. To adequately interpret this experiment, it is essential to know the kinetics of mRNA knockdown, GDF15 synthesis and secretion, and reduction on plasma GDF levels, results that are not provided. In addition, because the vector likely entered the circulation and not just the myocardium, systemic effects may have occurred, including knockdown of GDF15 synthesis in other tissues. No assessment of this possibility is provided.

Response: We agree with the reviewer that a kinetics study of GDF15 shRNA knockdown *in vivo* would be informative. We respectfully point out that a huge number of mice will be needed for such a kinetics study which makes it practically impossible to complete within the time allowed for the revision. To put this into perspective, it took us more than 6 months to complete such a study of one time point (10 days). For this reason we prioritized available $ERR\alpha/\gamma$ double KO mice for other critical studies suggested by the reviewers, including those addressing the 3rd and 4th points above which required many litters of mice and effectively exhausted all the mice we had during the revision period. We hope that this is acceptable.

We designed several strategies to ensure the cardiac-specific knockdown of Gdf15. First, because Gdf15 is exclusively expressed in $\alpha KO\gamma KO$ mouse cardiomyocytes, we designed the AAV vector to ensure that Gdf15 shRNA is solely expressed in Cre⁺ cells ($\alpha KO\gamma KO$ mouse cardiomyocytes, Fig 5A). Second, we used AAV9 serotype which was previously shown to achieve stable and relatively cardiac-specific expression of transgenes (<http://www.ncbi.nlm.nih.gov/pubmed/18795839>). Last, we directly measured Gdf15 expression in non-cardiac tissues including liver and subcutaneous fat as suggested by the reviewer, and observed no change (new Fig EV4D). These new data are included in the revised manuscript.

A further weakness of this experiment is that details of "intrapericardial" injection, a technically demanding or impossible task, and the success rate for actual knockdown are not given because all animals with "unsuccessful" knockdown are excluded. These data should be given.

Response: We thank the reviewer for this comment. We largely followed procedures previously described for mouse pericardial injection and have cited this paper which provides detailed

illustrations of the procedures. (<https://www.ncbi.nlm.nih.gov/pubmed/23250337>), adapting the technique to young mice without using ultrasound guidance. As quality control and based upon pre-established criteria, mice dead before 9 days of age or with unsuccessful cardiac Gdf15 knockdown (presumably due to unsuccessful injection or ineffective AAV infection) were excluded from the analysis. With these criteria, 2 of 10 α KO γ KO mice that received AAV9-shGDF15 and survived to 9-10 days of age in the experiment in Fig 5 were excluded from the study (one mouse showed little change in cardiac Gdf15 expression; the other mouse has been losing weight since 6 days of age and looked very sick/dying by day 9). Similar success rate was observed in control groups. These details are now included in the Materials and Methods section of the revised manuscript.

The results in children with "heart disease" are problematic, given the lack of any description of the severity of heart disease and the huge scatter in the results, with several outliers that skew the results. The means, medians, and standard deviations are not given, and the figure does not provide the number or percentage of patients with values above two standard deviations. In fact, even the vast majority of patients with poor growth overlap with normal. That is, only 7/45 appear to exceed the normal range. This might suggest that, in fact, elevated plasma GDF15 is not associated with poor growth. Review of the ICD9 codes suggest that almost all have congenital heart disease (CHD) (codes 745, 746, 747) and not cardiomyopathy (425.4, only 3 patients among 70 total) and that the entire spectrum of anomalies, from trivial to life-threatening is included. Cardiac patients with other causes of FTT, such as chromosomal anomalies which are common in CHD, are not excluded. Therefore, the conclusion that GDF15 levels may be a useful biomarker in children cannot be justified by these data.

Response: We appreciate the reviewer's comment. We have included the means, medians, standard deviations and % of patients outside of 2 standard deviations in a revised Table EV4 as requested. When we analyze data excluding samples that fall outside of 2 standard deviations as the reviewer suggested, the conclusion remains that children with heart disease and FTT have statistically higher plasma GDF15 than controls or heart disease and normal body weight. In fact, the difference becomes more statistically significant (the numbers are shown below). We removed descriptions of GDF15 as a potential biomarker following the reviewer's suggestion.

The discussion contains some inappropriate comments and does not focus on the key points. For example, the statement (page 12, lines 2-3), "However, pediatric heart disease results in decreased cardiac function that fails to match these increased demands" is far too broad and factually incorrect. Actually, most pediatric heart disease, especially CHD, does not cause symptoms and does not have decreased cardiac function/output.

Response: We thank the reviewer for this comment and have removed this sentence in the revised manuscript.

All samples

GDF15 (pg/ml)	Control	HD normal BW	HD FTT
mean	176	304	538
median	148	216	357
Standard deviation (SD)	108	300	560
Standard error of the mean (SEM)	16	52	84
% of samples outside of 2 SD	4.5%	5.9%	4.5%
t-test (control vs HD normal BW)	0.02		
t-test (control vs HD FTT)	0.00012		
t-test (HD normal BW vs HD FTT)	0.02		

Exclude samples outside of 2 SD

GDF15 (pg/ml)	Control	HD normal BW	HD FTT
mean	159	236	445
median	145	201	335
Standard deviation (SD)	73	127	318
Standard error of the mean (SEM)	11	23	49
t-test (control vs HD normal BW)	0.002		
t-test (control vs HD FTT)	0.0000007		
t-test (HD normal BW vs HD FTT)	0.0002		

Similarly, the statement (page 13, last two lines) "This is in contrast to most other heart disease animal models whose late-onset nature or early embryonic/neonatal lethality prohibited the chance to study the pediatric period." Is also incorrect. There are many mouse models that would allow examination in this same time frame.

Response: We thank the reviewer for the comment and have removed this sentence in our revised manuscript.

These limitations detract from the key conclusions, which are reasonably well supported. First, GDF15 is synthesized in the heart, induced in this mouse model, and does serve an endocrine role to signal hepatocytes. Second, this pathway may play a role in overall growth. These are important and exciting.

Response: We thank the reviewer for his/her positive comments that our studies are important and exciting.

However, the data do not support the authors' conclusion that cardiac GDF15 is the only factor altering growth in the mouse model and certainly not in children with heart disease.

Response: We thank the reviewer for this comment and have expanded in discussion about other potential mechanisms (page 15, highlighted):

Specific comments:

1. Introduction, page 3, line 6. A key endocrine organ to add to the list is the intestine.

Response: We thank the reviewer for pointing out this oversight. We have added intestine with relevant reference in our revised manuscript (page 3, highlighted).

2. Figure EV 1A, a "cartoon" of growth hormone signaling is superfluous, but could be part of a concluding figure to emphasize role of GDF15 in growth, e.g. Figure 6B

Response: We removed the cartoon (original Fig EV1A) in our revised manuscript following the reviewer's suggestion.

3. The key component of the data is hepatic phosphorylated stat-5 levels, as shown in figure 1E (10 day old) and Figure EV1, panel E (16 days).

Response: We made every effort to reorganize Fig EV1 and it now contains only essential data or those requested by other reviewers.

4. All other panels in Figure EV1 are adequately described in the text and can be deleted.

Response: We removed as much as possible the other data (except those requested by other reviewers).

2nd Editorial Decision

03 May 2017

Thank you for the submission of your revised manuscript to EMBO Molecular Medicine. We have now received the enclosed reports from the referees that were asked to re-assess it. As you will see the reviewers are now globally supportive and I am pleased to inform you that we will be able to accept your manuscript pending the following final editorial amendments:

1) Please carefully check the authors guidelines for formatting your supplemental information: Expanded view and/or Appendix (see: <http://embomolmed.embopress.org/authorguide#expandedview>)

You provided a single pdf file including 4 supplemental figures and 3 supplemental tables. The simpler option would be to relabeling all as "Appendix" and within the Appendix, add a Table of Content as the 1st page, then "Appendix Figure S1" and so on, "Appendix Table S1" and so on.

Please do not forget to update the callouts in the main article file.

2) GEO accession number: please make ensure that the data is publicly available upon acceptance of the article (it is now private until 2019)

We are looking forward to receiving the revised article within 2 weeks.

***** Reviewer's comments *****

Referee #1 (Comments on Novelty/Model System):

Authors have made an effort to answer questions raised by myself and reviewer 3 (which I was asked to take a look). I think their responses are very reasonable and appropriate. I don't have additional comments.

Referee #1 (Remarks):

The authors have addressed my comments. Majority of reviewer 3' concerns have also been addressed with new experiments and/or revised data interpretation.

Referee #2 (Remarks):

The authors have adequately responded to all my previous comments. As requested by the editorial office, I also had a look at their responses to reviewer #3; here, they also responded appropriately. I feel that this manuscript can now be published.

Corresponding Author Name: Liming Pei

Manuscript Number: EMM-2017-07604